# An Recognition–Verification Mechanism for Real-Time Chinese Sign Language Recognition Based on Multi-Information Fusion

**DOI:** 10.3390/s19112495

**Published:** 2019-05-31

**Authors:** Fei Wang, Shusen Zhao, Xingqun Zhou, Chen Li, Mingyao Li, Zhen Zeng

**Affiliations:** 1Faculty of Robot Science and Engineering, Northeastern University, Shenyang 110819, China; 1700972@stu.neu.edu.cn (S.Z.); 1801929@stu.neu.edu.cn (C.L.); 1801930@stu.neu.edu.cn (M.L.); 2College of Information Science and Engineering, Northeastern University, Shenyang 110004, China; 1770748@stu.neu.edu.cn (X.Z.); 20154557@stu.neu.edu.cn (Z.Z.)

**Keywords:** sign language recognition (SLR), recognition–verification mechanism, surface electromyography (sEMG), CNN, Siamese network, VGG

## Abstract

For online sign language recognition (SLR) based on inertial measurement unit (IMU) and a surface electromyography (sEMG) sensor, achieving high-accuracy is a major challenge. The traditional method for that is the segmentation–recognition mechanism, which has two key challenges: (1) it is difficult to design a highly robust segmentation method for online data with inconspicuous segmentation information; and (2) the diversity of input data will increase the burden of the classification. The recognition–verification mechanism was proposed to improve the performance of online SLR. In the recognition stage, we used sliding windows to pull the data, and applied a convolutional neural network (CNN) to classify the sign language signal. In the verification stage, the confidence was evaluated by the Siamese network to judge the correctness of the classification results. The accuracy and rapidity of the classification model were discussed for 86 categories of Chinese sign language. In the experiments for online SLR, the superiority of the recognition–verification mechanism compared to the segmentation–recognition mechanism was verified.

## 1. Introduction

Sign language recognition (SLR) refers to the process of classifying sign language by analyzing the motion information of the hand and arm acquired by the sensors. As a main way of communication between deaf people, sign language is widely used in hearing-impaired groups. However, most people of normal hearing generally do not understand sign language, so SLR has become an important research subject to help people communicate and live. In addition, as a special case of gesture recognition and human–computer interaction (HCI), sign language is more complex and changeable. Therefore, research on SLR can also greatly promote the development of gesture recognition and human–computer interaction. The recognition of Chinese natural sign language is a very complicated process. Natural sign language consists of elements such as hand shape, position, and movement. Unlike American sign language based on the letter-related hand shape, there are 61 types of hand shapes in Chinese natural sign language, which is twice that of American sign language. In addition, there are more than 8000 basic words in Chinese natural sign language, which greatly increases the complexity of SLR. In SLR, the difference between sign language made by different subjects is huge, which also causes a challenge for the recognition of Chinese natural sign language.

There are three main types of SLR, which are based on computer vision, data gloves, and other wearable devices. Compared to vision-based and data glove-based methods, wearable devices containing surface electromyography (sEMG) sensors and IMU have been widely used in the study of SLR [1,2] due to their advantages of low cost, low environmental dependency, natural gestures friendliness, good wearability, etc. However, the current research on SLR mainly focuses on optimizing the extracted features and improving the SLR algorithm to increase the accuracy rate [3,4], while ignoring the processing problem of continuous sign language signals in online recognition. Since online SLR is often accompanied by the interference of non-effective action information [5], those classification methods which perform high-precision in offline SLR may not do well in online recognition.

In recent years, the Myo armband, as a new type of wearable device, has been applied to gesture recognition and HCI. Fernando et al. [6] used the combination of the Myo armband and Leap Motion to estimate continuous hand position using the recurrent neural network (RNN). Li et al. [7] used the Myo armband to obtain the sEMG signal of the arm position and combined deep learning with principal component analysis (PCA) to control the grasp of a prosthetic hand; Ding et al. [8] combined a three-dimensional imaging sensor and wearable Myo armband device for HCI with a mobile service robot platform. Due to the good portability and anti-interference ability of the Myo armband, we applied it to our sign language recognition research.

How to recognize continuous sign language with higher precision from a long string of data whose segmentation information is not obvious is a core problem in SLR. Previous studies have been more inclined to solve this problem by using a segmentation–recognition mechanism. For example, in reference [9], a method based on the segmentation probability for sign language segmentation and recognition was proposed. In reference [10], the sEMG threshold was used to segment effective sign language signals. The endpoint detection method based on dynamic time warping (DTW) was used in references [11,12]. However, there is interference of non-sign language actions during the action of sign language, which makes the segmentation information inconspicuous. In the face of interference, it is difficult to determine the start and end positions of a sign using the traditional segmentation–recognition mechanism to accurately separate the valid part and the invalid part. The diversity of the input data will increase the burden of the classification model. Otherwise, the over-fitting inevitably occurs in the classification model, which makes the reliability of subsequent recognition results difficult to guarantee.

To solve the above problems, we used a wearable device integrated with sEMG and an inertial sensor for sign language data collection, and proposed a recognition–verification mechanism to realize online SLR. There are three main points in our contribution: (1) the recognition–verification mechanism is first proposed to complete online SLR, which overcomes the problem of extracting effective action patterns from continuous sign language data; (2) we are the first to use a convolutional neutral network (CNN) based on VGG to construct the classification model of sign language and migrate the convolution part of the classification model to the verification part. By optimizing the network structure, the network training overhead is saved while ensuring the accuracy rate; (3) we apply the Siamese network structure to the verification part of pattern recognition for the first time and reduce the probability of overfitting after online continuous signal recognition.

## 2. Online Sign Language Recognition

This section mainly analyzes the difficulties encountered in online SLR and the solution to traditional continuous signal recognition. Except for the effective sign language information, there are also some non-sign-language pieces in the input of real-time continuous signals which are not present in the offline recognition system but online. These non-sign-language pieces increase the difficulty and complexity of online SLR. The accuracy rate in online SLR is lower than that in offline. In the traditional continuous signal recognition method, the segmentation–recognition mechanism is used to separate the effective sign language segments. This method solves the problem of continuous signal recognition to some extent. However, the effect of this traditional method in online recognition still faces enormous challenges.

### 2.1. Analysis of Online Sign Language Recognition

In the online SLR, the use of the sEMG signal for sign language segmentation or sign language data segment detection is a common means [9,10,11,12]. However, the effective segmentation information of the real-time input data is not obvious. Figure 1 presents sEMG signals in the case that two kinds of sign language executed continuously. We can see that there are some non-sign-language signals between “today” and “morning”. In reality, the boundary waveform of non-sign language is very similar to those of adjacent sign language, and it is difficult to find the boundary between sign language signals and non-sign language signals through a general algorithm. 

We call the factors that cause the inconspicuous segment information interference. The causes of interference are different. Noise and non-sign language actions are the main sources of interference. Noise is an unavoidable interference factor in the process of signal acquisition. When the collected signal is weak, the signal-to-noise ratio of the signal is low. Noise will become a very strong interference factor. Usually, this situation will occur at the beginning and the end of the sign language action. It is difficult to deal with the low signal-to-noise ratio signal well by denoising, which will cause the accurate segmentation of the sign language signal. Non-sign language is another important interference factor. At the beginning and end of sign language action execution, additional non-sign language actions are usually accompanied. In addition, when people do not make sign language, they may also perform other actions, such as walking, swinging their arms, grabbing or raising their hands. When these actions are executed, the collected sign language signals are very strong, because they are not noise, but real body movements. These actions are complex, changeable, and difficult to predict in advance, so it is difficult to recognize these actions effectively. Therefore, the interference of non-sign language actions has a great impact on the process of sign language segmentation.

In order to effectively distinguish the influence of interference information in the online recognition process, we partition the collected real-time signal data segments, wherein the effective sign language signal segment is called the “effective region” and the signal segment between the effective sign language data is called the “blank region”, as shown in Figure 1. The blank region can be subdivided into two zones, wherein the data segment away from the effective region is called the “free zone”, and the data immediately adjacent to the effective region is called the “adjacent zone”. Regardless of whether the interference is in the free zone or the adjacent zone, the online SLR causes different degrees of influence.

We call the successful recognition of sign language data in the effective region “correct”, and the others misrecognition cases. In order to understand and distinguish the misrecognition cases in online SLR, we divide the situation of misrecognition into five categories, as shown in Figure 2:
Insert: the misrecognition that the data segment in the blank region is identified as an exact category of sign language.Misalignment: the misrecognition of the data segment combining the adjacent zone and part of the effective region data into a certain type of sign language.Repeat: the misrecognition that the same effective region is correctly recognized multiple times.Substitute: the misrecognition of the data segment of a certain type of sign language in the effective region into another type.Delete: the misrecognition that the entire effective region is recognized as the blank region.

The reasons for the above misrecognitions are different. When the non-sign-language action exists in the free zone, the signal presenting in this zone is more active, which may easily lead to misrecognition of “insert”. When the non-sign-language action exists in the adjacent zone, it is not easy to find the starting and ending positions of the effective sign language data in the online recognition, which leads to the occurrence of “misalignment” errors. However, due to the influence of subject-independent, the standard of action and other factors in the online recognition process, the input effective sign language data is diverse, and there may be differences from the sign language data collected during the original training process, which leads to the occurrence of “substitute” and “delete” errors. Since the classification model is robust, the same effective region was correctly identified multiple times, thereby causing “repeat” errors.

### 2.2. Traditional Segmentation–Recognition Mechanism

In the traditional online SLR, the segmentation–recognition mechanism, as Figure 3 shows, was used to extract the effective action from the continuous signal [9,10,11,12].

In the segmentation–recognition mechanism, the segmentation method was first designed to find the start and end points of the effective sign language signal from the continuous input signal, and then used the classification model to recognize the class of the valid sign language segment. This SLR mechanism can help extract effective sign language fragments in continuous signals and construct sign language datasets, which can help improve the accuracy rate of SLR in the recognition process.

Although the traditional segmentation–recognition mechanism provides a reference solution for the recognition of continuous sign language signals, the application of this mechanism to online recognition still faces a big challenge. In online SLR, the interference of non-sign-language action has an important impact on the accuracy rate. Such as the following cases: if there is an interference action in the blank region, misrecognition will occur because the signal strength of the region is strong, and it may be segmented as a valid segment during the segmentation process. After entering the classification model, if it was recognized as an action, an error of “insert” occurs. In another case, when interference appears in the adjacent zone, the segmentation–recognition mechanism finds it very difficult to determine the true sign language effective segment, since the action in this zone is a coherent action formed by the standard sign language action and the signal strength is strong. If this coherent signal was passed to the classification model during the segmentation process, the error of “misalignment” will occur. If the complete sign language action and the additional action are combined in the segmentation process, passing as an input signal to the classification model may result in a “substitute” or “delete” error. Besides, if only a portion of the valid sign language action signal is transmitted to the classification model, it may also result in a “delete” or “substitute” error.

In the process of online SLR, the effective sign language data segment of the continuous data transmitted is not very obvious due to the interference action, which makes it difficult to design an efficient segmentation method to accurately segment complex continuous data. The method of denoising can effectively eliminate the noise interference in online recognition, but it is ineffective for non-sign-language action interference. In addition, due to the diversity and complexity of the data segment input to classification model, the burden of the classification model is increased, and the classification model inevitably has a certain over-fitting characteristic, resulting in an increase in the classification error rate. In order to extract real-time continuous signals and recognize effective action modes, it is necessary to create a new mechanism to realize online SLR.

## 3. Database

Since the specific data processing method needs to be made according to the input data in the SLR’s stage, before describing the recognition–verification mechanism we created, we first introduce the sign language signals and the sign language database we collected.

### 3.1. Sign Language Signals

In this paper, we use sign language signals for SLR collected by wearable devices that include sEMG sensors and IMU. Compared with the methods of using computer vision [13,14], this method is not affected by factors such as light intensity, background color, and motion occlusion, and there is no need to attach the device to somewhere outside the body in order to ensure that the sign language information of the object can be obtained at any time, so it is more portable. Compared to data gloves [15,16], this method is more convenient to wear and does not affect the naturalness of action.

The wearable device shown in Figure 4 that we used to collect the sign language signal was fixed at the muscle group position of the subject’s arm. The device integrated an 8-channel sEMG sensor and a 6-channel IMU. The sEMG sensor was used to collect muscles’ electrical signals in the generated gestures, and the IMU included 3-axis accelerometer (ACC) and 3-axis gyroscope (GYR) to capture motion information in gestures. 

Figure 5 shows six classes of sign language. In the process of analyzing different kinds of signals, we will discuss the similarity and difference of different classes of sign language signals in combination with Figure 5. The signal of sEMG is an electrical signal generated by the potentials generated through the contraction of many fiber bundles in the muscles, which are superimposed in time and space, such as in Figure 6. The signal has strong volatility and instability [17]. In Figure 7, we can see that the fluctuations of sEMG signals in different sign languages are different. Compared with Figure 7a–d, we can see that the sign “luggage” has the largest scope of forearm movement, so the activity of the forearm muscle of this action is the most intense. Therefore, the sEMG signal of the sign “luggage” has the largest amplitude. The highest frequency information was in the signal of the sign “thanks” and the signal of the sign “friend” had the lowest frequency information. Because the action of signing “thanks” can cause intense activity of the thumb extensor and the activity scope of signing “friend” the smallest. Through analyzing the fluctuation characteristics of sEMG generated by related muscles during the sign language execution, we can achieve the purpose of SLR.

The acceleration information collected by the IMU is a kind of motion information, including ACCs and GYRs [18]. In the process of sign language execution, the movement of the arm will cause the change of linear acceleration and angular acceleration at the position of the sensor. Figure 8a,b represents the ACC curves of the horizontal axis. In the horizontal direction, the forearm of the sign “communication” is rotated, so the ACC signal shows regular fluctuation. The horizontal direction of the sign “you” is a forward movement, so the ACC signal has an obvious peak. The GYR curves in the vertical direction are plotted in Figure 8c,d. The GYR curves of the sign “communication” and “you” are similar because of their similar vertical swing arm movements. By capturing these acceleration changes with motion sensors and processing and analyzing them with a pattern recognition method, the types of sign language actions in the process of sign language execution can be distinguished.

Although both signals collected by the sEMG sensor and the IMU can be used to recognize sign language, the characteristics of the two signals are different. The inertial information collected by the IMU can effectively recognize the motion category of the forearm, but not of the wrist and the palm of the hand. The movements of the hands and fingers, as well as the strength information in the sign are difficult to recognize, while the sEMG signals can efficiently describe this information to achieve good recognition [19]. The combination of the two sensors to jointly collect the relevant signals in the sign language execution process can better complete the SLR work.

### 3.2. Sign Language Database

Although the work of SLR using sEMG and inertial signals has been studied, this research is still a relatively new field. There is no existing database to provide reference and research [20]. Therefore, in order to study and implement an effective scheme of online SLR and test the scheme, we created a sign language database based on sEMG and inertial signals collected by ourselves.

Sign language is a kind of communication method with high flexibility and variability. In order to cover the sample space of sign language as much as possible, we divided the data collection work into multiple groups. Each group of data contains one or more categories of sign language, and each category of sign language contains 20 samples. In addition, the sign language actions made by different subjects are less standardized, so the subjects need to learn sign language systematically before data collection.

To overcome non-specific issues, we invited more than 20 subjects between the ages of 20 to 26 to collect data. Each subject was healthy in the right hand and had no history of neuromuscular disease. The energy distribution of the sEMG signal was in the range of 20–500 Hz, and the sEMG signal at 50–200 Hz was most excellent in SLR [21]. The acquisition frequency of the signal was set to 100 Hz. During data collection, each subject chose several categories of sign language for acquisition. Each sign language was collected in groups with 20 samples each time. In order to ensure the quality, we conducted a manual screening for the sign language data collected.

As shown in Table 1, the sign language database we built contains 86 categories of Chinese natural sign language with empty classes. The data size of sign language data ranges from 164 to 207 points. In the model training phase, we extracted 128 data points from the core of the sign language samples for pre-processing. After pre-processing, the sign language vector with 64 dimensions was transmitted to the classification model and verification model. When the “empty” category is not included, the maximum number of single sign language samples is 2964, and the minimum is 564. The dataset of the “empty” category contains 361 samples. The database contains 85,424 samples, wherein the dataset of each category contains an average of 993.30 samples.

## 4. Method

In order to solve the difficulties encountered in online SLR, different from the traditional segmentation–recognition mechanism, we propose the recognition–verification mechanism. The mechanism is divided into three parts: data preprocessing, the classification model, and the verification model. These three parts are connected in an assembly line that continuously pulls data from the data source and processes the sequential signal of the online input from sensors in real time to output the effective result. 

### 4.1. Recognition–Verification Mechanism

In view of the difficulties encountered in online SLR, the recognition–verification mechanism is proposed as shown in Figure 9. Traditional SLR is completed by the segmentation–recognition mechanism, where the segmentation algorithm was designed to segment effective signal segments of sign language from the real-time input signal and then the classification model is used to recognize them. The effect of this mechanism depends on the accuracy rate of segmentation and the robustness of the classification model. Due to the existence of non-sign-language actions and noise interference in online sign language data, it is difficult to guarantee the segmentation’s effectiveness. When the non-sign-language signal is passed into the classification model, the output of the recognition result will also present an uncertain state, which leads to the great increase in misrecognition in online recognition.

In the recognition–verification mechanism, the traditional segmentation operation is discarded. The data fed into the mechanism is continuously pulled from the data source by sliding window in advance, and the data is transmitted to the classification model in real time. The length of the sliding window is set to 128 to pull the core part of the sign language data, and the sliding step is set to random integers from 8 to 20. The working principle of the sliding window is shown in Figure 10.

In order to describe the principle of pulling data of the sliding window, the change of the sliding window position in the Figure 9 replaces the real data pulling situation. In the process of online SLR, corresponding to the form of the sliding window moving, fixing the window in one place, once the input data fills the sliding window, the system begins to pull the set of data and pass it into the recognition–verification mechanism to complete recognition and verification. At the same time, the sliding window slides forward one step. When the new input data fills the sliding window again, the same process is performed until the recognition process is at the end.

Instead of the segmenting operation, the sliding window is used to pull the data, which results in the amount of data passed into the mechanism being larger and the diversity of the data greatly increasing. In order to ensure a high accuracy rate, we added a verification model to judge the correctness of recognition of the classification model. 

The classification model is the core of the whole mechanism. When the sign language samples pulled are sent to the mechanism, the data are sent to the classification model for preliminary recognition after data pre-processing. It is worth noting that, without the segmentation process, there is no corresponding operation to select the effective region from the input raw data, so that the input data are no longer mainly from the effective region, but may come from a combination region of blank region and effective region data. We call the data coming mainly from the effective region as “benign data”. They are mainly composed of data of the effective actions of sign language, which the classification model can easily recognize as the correct result. Other data are called “malignant data”. Unfortunately, the amount of “malignant data” entering the classification model far exceeds that of “benign data”. The input of “malignant data” will cause misrecognition of produce, and a large number of “malignant data” will lead to extremely chaotic classification results. Increasing the robustness of the classification model can improve the recognition effectiveness to some extent. However, in the face of such a large number of “malignant data”, it is difficult to say that the solution is better.

The function of the verification model is to verify the correctness of the recognition results of classification models by calculating the difference. Its working principle is to use the trained verification model to generate the standard reference vector of each category of sign language in the model training stage. In the online recognition stage, the verification model encodes the data pulled through the sliding window into the current sign language vector the same way. The mechanism selects the standard reference vector from the set of vectors of all categories based on the output category of the classification model. Then, the difference calculated from the current sign language vector and standard reference vector is used to judge the recognition result. The “benign data” and the samples of the same category in the database are relatively similar, and the difference between the generated encoding vector and standard reference vector is accordingly small; they are more likely to be recognized successfully. Even if the “malignant data” is misrecognized, the difference between the generated encoding vector and standard reference vector is generally large. The verification model can effectively filter out the misrecognition results of the classification model.

### 4.2. Data Pre-Processing

In order to get better recognition results, the data was pre-processed before it was transferred into the classification model. Pre-processing was used to eliminate the interference of the raw data. Pre-processing performs the same operations in model training and online recognition. In the model training phase, we extracted 128 data points from the core of sign language samples for pre-processing. In the online recognition stage, data pre-processing was directly used in the data of 128 lengths pulled by sliding window. Regarding the signals collected by different sensors, we adopted different data processing schemes due to the different characteristics of the signals.

The IMU was used to collect traditional Newtonian physical signals in sign language actions, including 3-axis ACC signals and 3-axis GYR signals. Through research, we found that there was a large amount of jitter in the collected data during the execution of sign language actions. In order to eliminate the interference of such jitter on the classification effect, we adopted the method of polynomial fitting combined with down-sampling to process the signals collected by the IMU. This processing method is a means to smooth the data, eliminate meaningless noise and jitter, and to only retain effective information in the data body. In the data fitting stage, according to the transformation frequency of data, we used the quadratic polynomial with 16 data points as a unit for fitting, and the polynomial is as follows:(1)$$y\left(x_{n},\omega \right)=Ax_{n}{}^{2}+Bx_{n}+C$$
where ω=(A,B,C), A, B, and C are the fitting coefficients, $x_{n}$ is the input value of the nth point, n is a serial of numbers ranging from 1 to 16, and $y\left(x_{n},\omega \right)$ is the output value about $x_{n}$ and ω. We used the sum of the squares of the errors as the loss function, and its formula was as follows:(2)$$L_{2}\left(\omega \right)=\sum_{n=1}^{16}{\left(y\left(x_{n},\omega \right)-y_{n}\right)}^{2}$$
where $L_{2}\left(\omega \right)$ is the loss value about ω. By minimizing the value of loss function, the coefficients of the polynomial were calculated. In the stage of data restoration, the polynomial coefficient obtained was put into Formula (2) to calculate the restored data at the corresponding position. The data was recovered at every other data point via down-sampling to replace the original data so as to achieve the purpose of data smoothing and compression. However, the piecewise fitting method is a regional fitting method, which will lead to data discontinuity and data jump in the connected region. In order to solve this problem, we improved the polynomial fitting: by setting-up the overlapping part when fitting the section between two units, and in the process of restoration, we calculated the average of the overlapping part to effectively eliminate jump. The data smoothing effect is shown in Figure 11. After data smoothing and down-sampling, the data size of sample becomes 64.

The signal of sEMG is a kind of bioelectrical signal, which is similar to an electroencephalogram (EEG). Referring to reference [22], we found that the frequency domain of noise signal is less than 15 Hz, and when the target muscle is active, its amplitude will change greatly [23]. Therefore, we firstly filter the signal in the frequency domain and the amplitude threshold to preliminarily eliminate the noise. Subsequently, we used the discrete wavelet transform (DWT), which is a signal processing method successfully applied on the unstable signal [24]. The general formula of the DWT is as follows:(3)$$W_{f}\left(j,k\right)=\int_{-\infty }^{+\infty }f\left(t\right){\bar{\psi }}_{j,k}\left(t\right)dt$$
where f(t) is the original sEMG signal, j and k are integers related to discretization of scaling and displacement parameters, ${\bar{\psi }}_{j,k}\left(t\right)$ is discrete basis vector, $W_{f}\left(j,k\right)$ is the transformation result of original signal f(t). 

According to Reference [22], the signal process of wavelet transform is to select a wavelet family and its transform layers according to the frequency of the input signal, signal granularity, and signal-to-noise ratio to achieve the best effect. With reference [25], we finally determined to apply the db5 wavelet in the Daubechies wavelet family to do 4 layers of multilayer wavelet transform for the data to get the waveform characteristics of the timing signal. In order to obtain the same length as the data collected by the IMU, we extended the transformed sEMG data. The processing effect of sEMG signal is shown in Figure 12.

### 4.3. Classification Model

#### 4.3.1. 1D Convolutional Neural Network

The nature of SLR is multi-classification for a period of a 1D time-series signal. Studying the data form, we can find that data have strong local correlation, thus we adopted a 1D convolutional neural network (1D CNN) algorithm for the classification model of sign language. The CNN was first applied in the field of computer vision in the case of large data volume and multi-classification [26,27,28,29] and achieved great success. The sharing of convolution kernel parameters in the hidden layers of a CNN and the sparsity of the connection between layers enables the CNN to learn lattice features with a small number of parameters. Compared with the traditional manual feature extraction method, this method has a stable effect and no additional feature engineering requirements for data.

In this paper, we used the VGG architecture to build a CNN model for SLR. The model of VGGNet was first applied to large-scale image recognition problems. The model obtains a larger receptive field by increasing the depth of the network and using a very small (3 × 3) convolution filter. Compared with the traditional CNN network, the model parameters were reduced, and the results were better than the traditional network of a large convolution core and less layers [30]. In addition, this model had a good generalization effect for other datasets. We applied the VGG architecture, which was originally applied to the 2D image classification problem, to the 1D SLR problem.

The input signal of sign language consists of a 3-channel ACC signal, 3-channel GYR signal, and an 8-channel sEMG signal. The length of each channel was 64 after data pre-processing. Although the three types of signals have their own characteristics, they describe sign language in different forms, and the signal has time consistency. It was assumed in this paper that good feature extraction and classification effects can be achieved by directly transmitting the signals into CNN without using independent channels.

#### 4.3.2. Classification Model

In this paper, a 1D CNN based on VGG architecture was used to build the classification model. The input of the network was the timing data of sign language, and the output was the category mapping of the input sequence. The final network structure is shown in Figure 13. The network was composed of convolutional layers and fully connected layers. The convolutional layers were used to fuse different kinds of sign language signals and extract features. 

In the VGG architecture, feature extraction was accomplished by blocks composed of multiple convolution layers using small convolution kernel (3 × 3). In general, in a neural network, the layer far away from the input contains more information. In order to ensure that the deep network contains more complex characteristic information, the number of convolution layers in the block close to the output (behind block) should be no less than that in the block close to the input (front block). In the problem of SLR, front blocks are mainly responsible for the signal pre-processing and signal fusion, while behind blocks contain more of the depth features of sign language. The block structure is shown in Figure 14.

In each block, the convolutional layers and the activation function are combined to complete the non-linear description for sign language signals. A single block may contain multiple contents of such combination to obtain stronger non-linear expression ability of the sign language signal and better complete the feature extraction. Between the convolution layer and the activation function layer, we added a batch normalization (BN) layer to improve network performance and stability. The pooling layer is placed at the output position of each block for automatic filtering and data compression of the features extracted.

Between the parts of the convolution and full connection (FC) of the network, we adopted the adaptive average pooling layer [31]. By pooling a high-dimensional time-series vector, we only saved the information-efficient part, which reduced the dimension of the convolution output and effectively realize the function of feature selection. The method has been effectively used in human action recognition in video [32].

#### 4.3.3. Training

In this paper, we used a sign language database to collect and train the classification model. We took 10% of samples as the testing set and the rest as the training set. The datasets were passed into the network in batches, each batch containing 128 samples. Before the model training, we used the normal distribution method to initialize the weights of the classification model. This network used the Adam optimizer to optimize network parameters. The initial learning rate of the network was 0.0001, and the learning rate was adjusted by exponential attenuation. The learning rate decreased by 0.1 times every 10 epochs.

### 4.4. Verification Model

In the process of online SLR, we used the sliding window to process the continuous input data. In order to avoid the data segment of effective sign language being skipped in the process of window sliding, the stride should be far smaller than the window size. This increases the possibility of which “malignant data” passes into the classification model. Due to the limited number of training samples and the strong black box, the network is inevitably over-fitting, resulting in its limited ability to deal with the wrong input. The confidence obtained by the SoftMax function cannot guarantee the classification effectiveness. For example, if the input consisting from each half of the two samples is passed to the network, the model also outputs a high degree of confidence. Therefore, we propose to use the verification model to judge whether the classification result was accurate.

#### 4.4.1. Verification Model for Sign Language Recognition

We used the Siamese network based on VGG architecture to verify the classification results, and this model has been widely used in facial expression recognition and object tracking [25,33]. The structure of the Siamese network is shown in Figure 15. The Siamese network is used to measure the similarity of two samples: the input of the model is a pair of samples and the output is the distance or difference value of the two samples. If the categories of the two samples are same, the model tends to output a smaller value, otherwise it tends to output a larger value.

Two input samples of the model are first fed into two coding networks, which share the same structure and the same parameters. After passing through the coding network, the data of the two samples are mapped to a multi-dimensional Euclidean space and generate two coding vectors with 64 dimensions. These two vectors are passed into a distance function to calculate the distance between them. The encoding network is divided into two stages: feature extraction and feature encoding. In the feature extraction stage, we selected the same structure of the classification model. After being flattened, the size of the extracted feature was 1024. In the feature encoding stage, an FC, whose input size was 1024 and output size was 64, was used to encode the feature. For the measurement of coding vector distance, we used the Euclidean distance function to calculate.

#### 4.4.2. Training Method

Different from the classification model, the Siamese network used to verify the model contains one pair of samples at a time. The matching and extraction of sample data pairs needs to be completed before the model training. In the dataset of the Siamese network, data pairs of the same category and different category account for 50% [34]. In order to ensure the adequacy of data matching, the total number of data pairs we extracted was five times the total dataset of the sign language.

In the process of training, data pairs were sent to the network for parameter optimization by means of batch training. In order to improve the training efficiency, we designed a loss function based on Sigmoid function. Similar to the classification model, the Siamese network also used the Adam optimizer to optimize the network parameters. The initial learning rate of the network was 0.0001, and the learning rate was adjusted by exponential attenuation. The learning rate decreased by 0.1 times every 10 epochs.

The training of the Siamese network used 90 percent of the sign language pairs and the remaining 10 percent of the pairs were used for network testing. In order to obtain the network training situation, in addition to using the average loss value as the basis of the training stage, the statistical value of the test results was also used as the evaluation criteria. In the case of model convergence, the output distance of different data pairs increased gradually, while that of the same data pair decreased. We chose the 95% quantile (L1) of the output distance with different data and 5% quantile (L2) of the output distance with the same data as the benchmark of test. When L1 was much larger than L2, and the status tended to be stable, the training process of the verification model was completed.

#### 4.4.3. Operation Stage

In the use stage, the input data will be encoded into a coding vector by the coding network. Each sign language category retains a standard reference vector in advance. According to the classification results of the classification model, the input coding vector was compared with the standard reference vector of the corresponding category and the difference value was calculated. The correctness of the classification result was judged by comparing the difference value and the threshold value. If the difference value was less than the threshold value, the classification result was judged to be correct, and on the contrary, the result wrong. Because there were different distributions of difference values in different categories of sign language, the threshold was set to a value related to the category of sign language.

For the acquisition of reference vectors, we first encoded all types of training data to obtain a large number of coding vectors, and then calculated the average value of each class of vectors to obtain the standard reference vectors. For the threshold value, we used the reference vector as the basis to calculate the difference value with all the samples of the same type and obtained the difference value sequence. The ascending sequence $S_{d}$ was obtained by ascending the order of each kind of difference sequence. The threshold value of each class was 2.5 times of 80% quantile. The formula is as follows:(4)I=|0.8N|
(5)$$V_{Th}=2.5S_{d}\left(I\right)$$
where $S_{d}$ is the ascending sequence of the difference value of a single category, N is the number of elements in $S_{d}$, I is the ordinal number of 80% of the quantile, $S_{d}\left(I\right)$ is the element which order number is I in $S_{d}$, and $V_{Th}$ is the threshold of the category.

According to the comparison, only the recognition results with the difference value less than the threshold value were output. When a correct result was encountered, the sliding window of several subsequent steps was skipped, so that part of the correct data would not be calculated in the model, thus avoiding the waste of computing resources and avoiding repeated errors.

## 5. Experiments

This section introduces the main experiments involved in the recognition–verification mechanism, including the optimization experiments of the classification model and the verification model, and the online recognition experiments of the recognition–verification mechanism. Section 5.1 and Section 5.2 aim to analyze some parameter options of the classification model and verification model. The final parameters of these two models are given in Section 5.3.1.

### 5.1. The Optimization of the Classification Model

In order to improve the accuracy rate of the classification model and speed up the network training and operation, we adjusted and optimized the parameters of the classification model. The following describes the optimization methods one by one.

#### 5.1.1. The Layers of the Network

On the basis of the VGG architecture, we refer to the VGG network structure given in reference [30] to adjust the number of blocks in the network and the number of layers in the block to optimize the network. The main structure of the network built in the experiment is shown in Table 2: There are five structures in Table 2. Conv3 represents the set of the convolution layer, BN layer and activation function layer with the kernel size of three, and the number after the symbol “-“ represents the number of output channels. Each box in the table shows a block of layers, and the output of each block is followed by a “MaxPooling” layer. In the result analysis, the changes of loss and accuracy rate, training time, and testing time will be discussed. By comparing the performance of different network structures, a network structure with the best comprehensive performance is finally selected. 

#### 5.1.2. Adaptive Pooling Layer

After the last block of the CNN, we added an adaptive average pooling layer to optimize the network. After the multi-layer convolution operation, the output of the convolution part contains a large amount of depth information, which not only has a high dimension but has a large amount of redundant information. By adding an adaptive average pooling layer, the information in the signals will be automatically learned and screened, and the data dimension reduction will be achieved. This process effectively replaces the classification function of the FC functionally and reduces a large number of network parameters by reducing the number of FC, so as to avoid the phenomenon of network overfitting. In this optimization experiment, we fixed the output of the adaptive average pooling layer at two and the number of FC layers was reduced from three to one. By analyzing and discussing the same parameters as in Section 5.1.1, the experimental result is compared with adaptive average pooling layer or not.

#### 5.1.3. The Activation Function

The VGGNet proposed in reference [30] was applied in the field of image classification, and rectified linear unit (ReLU) was used as the activation function. In recent years, the Leaky ReLU activation function has been applied to the classification of CNNs, and its performance was analyzed in reference [35]. In reference [36], considering the trade-off between network sparsity and performance of the Leaky ReLU function, Leaky ReLU is used as an activation function to classify sound signals in a CNN network. The sign language signal processed in this paper belonged to a one-dimensional fluctuation signal as well as a sound signal. Referring to reference [36], Leaky ReLU is also used as an activation function in our classification model. The formula is as follows: (6)$$f\left(x\right)=\left\{ \begin{array}{l} x,x>0\\ \lambda x,x\leq 0 \end{array}\right.$$
where x is the input value of the activation function. f(x) is the output value. When x is negative, λ is the slope of the activation function. In this experiment, the value of lambda is set to 0.01. The advantages of Leaky ReLU over ReLU were analyzed by analyzing and discussing the same parameters as in Section 5.1.1.

#### 5.1.4. Batch Normalization

The layer of BN, as an important result of deep learning in recent years, has widely proved to be effective and important [37]. Its advantages are as follows:The layer of BN can alleviate the internal covariate shift, improve the stability in the network training process, and make it possible to use a large learning rate in the network training process, so as to accelerate the training of neural network.The problem of gradient explosion and disappearance can be alleviated to a certain extent, thus making it possible to train the deep network.Since the input of the hidden layer is processed in a standardized way, BN can make the network training process less affected by parameter initialization.

Therefore, this paper tries to add a BN layer into the network and compares the network training effect of adding BN layer or not by analyzing and discussing the same parameters as in Section 5.1.1.

### 5.2. The Optimization of the Verification Model

In order to improve the validating effect of the verification model and reduce the training time of the Siamese network, we optimized the verification model by experiment.

#### 5.2.1. Loss Function

The traditional Siamese network uses the contrastive loss function to estimate the performance of the model [38]. The input of this function is the distance of the coding vector generated by the pair of sign language data, and the output is the loss value of the pair of data processed by the network. The expression is as follows:(7)$$L\left(M,Y,X_{1},X_{2}\right)=\left(1-Y\right)\frac{1}{2}D_{M}^{2}+Y\frac{1}{2}\max{\left(0,m-D_{M}\right)}^{2}$$
where M refers to the model of the Siamese network after each iteration, $X_{1}$ and $X_{2}$ is the coding vectors of sample pairs, $D_{M}=D_{M}\left(X_{1},X_{2}\right)$ represent the Euclidean distance between the vector $X_{1}$ and $X_{2}$ under the model M; Y represents two vectors of category labels, which are the same or not, where “0” is on behalf of the same category, “1” represents different categories; m is the lower margin of distance desired when the category is not the same. When categories are different and Euclidean distance is greater than this threshold, the output of the L value is 0. When the categories are the same, the value of L increases with the increase of the input $D_{M}$ value; otherwise the value of L decreases, until $D_{M}$ is greater than the margin.

Employing the contrastive loss function can ensure that when the categories are the same, the output distance of the model tends to be a relatively small value, while when the categories are different, the output distance of the model is generally a relatively large value. However, when the function is used as a loss function to train twin neural networks, the sparsity of the network will be large. At the stage of model training, when different kinds of sign language data are transmitted to the network, the model output is larger than the margin, and the result of the gradient calculation is 0. The existence of the sparsity problem leads to the phenomenon of fluctuation and slow training speed in the process of training.

To avoid the sparsity problem, we created a new function as the loss function of the Siamese network. This function is based on the Sigmoid function, which is called the “sigmoid modification function”. Its mathematical formula is as follows:(8)$$L\left(M,Y,X_{1},X_{2}\right)=\frac{1}{1+e^{-D_{M}+m}}$$

We call the m a boundary point, and the remaining parameters have the same meaning as the contrastive loss function. The value of L is distributed between 0 and 1, which can be regarded as the probability of different categories of input data pairs. When larger than the margin, the value of L is greater than 0.5, more than 50% of the input data is likely to be of different categories.

The distance output of the Siamese network is not less than 0, within which the sigmoid modification function is a smooth curve. This function not only avoids the sparsity problem of contrastive loss function, but also has a large gradient at the position close to the boundary point in the process of training, which enables the verification model to have a better discrimination effect after the training convergence. The variation of loss and the frequency distribution of different distances will be discussed to prove the validity of our loss function.

#### 5.2.2. Model Initialization

Model initialization is a very important step in network training. The degree of model initialization will have a great influence on the speed and even the effect of model training [39]. Since parameters of the convolution part have been trained in the classification model, the CNN using these parameters has achieved a good classification effect and has also proved the feature extraction ability of the convolutional layer using these parameters. In this experiment, the characteristics of the network coding part adopts the normal distribution weights initialization method. The network structures of feature extraction are exactly same as the classification model’s, so we will transfer the parameters of the convolution part to the verification model to initialize the convolution part of the Siamese network. By analyzing and discussing the same parameters as in Section 5.2.1, we compare the training model with another initialization case which the whole network initializes weights by the normal distribution method.

#### 5.2.3. Batch Size

During model training, datasets are usually sent to the network in batches. Selecting the batch size is a very important task in the process of network training. When the batch size was too large, the number of iteration times of each epoch network was too small, resulting in slow training speed. Meanwhile, the normally large batch size would reduce the generalization ability of the network. Therefore, the batch size should not be too large when conducting network training [40]. However, when the batch size is too small to reflect the overall situation of the datasets, it results in excessive fluctuations in network training. In our experiment, we set the batch size to 128, 256, and 512, respectively, and then compared the three conditions to select the most suitable batch size for this model by analyzing and discussing the same parameters as in Section 5.2.1.

### 5.3. Online Sign Language Recognition

In order to verify the effectiveness of the recognition–verification mechanism, we designed an online sign language recognition experiment and compared it with the traditional segmentation–recognition mechanism.

#### 5.3.1. Online Dataset

For ensuring the consistency of data sources of the two mechanisms, a set of online sign language data was collected in advance as the validation set. Figure 16 shows a sign language signal intercepted from online data. 

The subject of the online dataset was involved in the construction of offline databases. The data we collected takes 2226 s and contains 20,383 data points. The dataset contains 437 effective sign language actions, including all kinds collected in the offline dataset. In the process of data acquisition, the time interval between two sign language actions is 0.5–5 s at random. At intervals, some non-sign-language actions of the subjects were recorded as interference in online data, such as walking, raising hands, grasping, swinging arms, or the combination of these actions. As can be seen from the figure, the strongest interference is the part between the signs “we” and “help”. Because the hand-raising and fist-clenching were carried out simultaneously, it is difficult to distinguish the starting point of the sign “help” from any of the three signals. Between the sign “help” and “you”, the interference of the sEMG signal was very strong due to the influence of arm force.

We passed the data of the online verification set into the two mechanisms and analyzed the performance of the two mechanisms by comparing the number of “insert”, “misalignment”, “repeat”, “substitute”, “delete”, and “correct”.

#### 5.3.2. Recognition–Verification Mechanism in Online Recognition

Through the optimization experiments of the classification model and the verification model, we selected some important parameters in the recognition–verification mechanism. The convolution part of the CNN uses structure D in Table 2 and we used the adaptive average pooling layer to reduce the number of layers of the full connection layer. The activation function of the classification model was Leaky ReLU, and the batch standardization was added to the model to optimize the training process of the model. In the verification model, we used structure D in the feature extraction, and we used the FC with an input size of 1024 and output size of 64 for feature coding. The length of the encoding vector was 64, and we calculated the Euclidean distance with the standard reference vector as the difference. The selection of thresholds is shown in Equations (4) and (5).

#### 5.3.3. Segmentation–Recognition Mechanism in Online Recognition

Reference [41] was used for the design of the recognition–verification mechanism. The main flow of the mechanism was divided into pre-processing, data segmentation, feature extraction, and sign language classification, and its flow chart is shown in Figure 3. However, in order to obtain a better online recognition effect, we made some modifications to several details and specific parameters.

In the pre-processing stage, the low-frequency noise in sEMG was eliminated by using a 5 Hz infinite impulse response (IIR) digital high-pass filter. The original signals were used in the ACC and GYR. Data segmentation used an average energy of multi-channel sEMG signals. A non-overlapping window contained 16 data points of sEMG to calculate energy. The calculation formula is as follows:(9)$$E=\frac{1}{n}\sum_{j=1}^{n}\sum_{c=1}^{m}R_{c}^{2}\left(j\right)$$
where $R_{c}\left(j\right)$ represents the jth data point of the cth channel of sEMG, m=8 is the number of sEMG channels, n=16 is the length of the window, and E is the energy value. In this experiment, the threshold of energy was set to 582. If four of the six consecutive windows were larger than the threshold, the first data point of the first window was regarded as the beginning of sign language. If five of the seven consecutive windows were below the threshold, the first data point of the first window was considered as the end of sign language. For feature extraction, sEMG uses four features of root mean square (RMS), zero-crossing (ZC), autoregressive coefficient (ARC) and short-time Fourier transform (STFT), and inertial information uses four features of RMS, ARC, power spectral density (PSD) and wavelet transform (WT). Finally, support vector machine (SVM) with radial basis function kernel is used for classification, which is based on “scikit-learn” in python. In SVM, the penalty parameter “C” is 256, the kernel coefficient “gamma” is 0.0175, and other parameters are default.

## 6. Results

In this section, we demonstrate and analyze the results of the parameter selection experiments of the classification model and verification model designed in the previous section. After discussing the results, some important parameters will be selected. In the online recognition experiments, the experimental results of the recognition–verification mechanism and segmentation–recognition mechanism will be analyzed. Finally, the experimental results of the recognition–verification mechanism and the traditional segmentation–recognition mechanism are compared.

### 6.1. Classification Model

The accuracy rate, training speed, and running speed of the model are important indicators to measure the quality of the classification model. The higher the accuracy rate is, the more reliable the result of SLR. Faster training speed can ensure that developers and users of the SLR system obtain a new model in less time. Faster running speed can allow us to obtain the results of recognition faster, and it is also a necessary condition to ensure online recognition. It is worth mentioning that batch processing of data can speed up the operation. Even though more samples are used in model training, each generation of training takes more computing time than each generation of testing. Because, in each operation, batch data is used in the training process and only one sample is used in the testing process. In the analysis of experimental results, we used these three indicators to compare the selection of different model parameters. The results of model parameter selection are given in Section 5.3.1.

#### 6.1.1. The Layers of Network

In the convolution part of the classification model, we built five network structures. The comparison results are shown in Figure 17. 

In order to describe the experimental results more clearly, Table 3 illustrates the experimental results. Wherein the “loss” label stands for the value of loss function, under which displays some parameter indicators when the specified value was first reached. “e” stands for epoch, “t” stands for time(s), and “AR” stands for accurate rate. In VGGA, when loss = 0.01, the three indicators are 0, indicating that loss does not decrease to 0.01 until the end of model training.

It can be seen from Table 3 that when the loss is at the same level, there is no significant difference in the accuracy rate of the classification models with different structures. Therefore, the loss value and accuracy rate can be regarded as equivalent to the performance of the model. Combining Table 2 and Table 3, and Figure 17, it can be seen that with the increase in the number of convolution layers, the training time and test time of each generation are slightly increased. However, when the number of layers remains unchanged, the change in the number of blocks has little effect on the training time and testing time of each generation. Therefore, the number of convolution layers determines the computing time of the network.

When using a CNN to process sign language signals for SLR, the convolution part is used to extract features, and the FC part is used for feature classification. In VGG architecture, each block operation extracts deeper features from sign language data, and the number of layers in the block determines the complexity of the features that the block can extract.

By comparing the structures of A, D, and E, extracting deeper sign language features by increasing the number of blocks can help the model achieve stable convergence with fewer iterations. However, as the number of blocks increases to a certain extent, depth features can accurately represent the characteristics of sign language data, and the acceleration effect of extracting deeper features on model training will gradually weaken. If the extracted sign language features are too deep, the training speed starts to slow down, since the increase in the number of blocks makes each iteration take more time. By comparing the combination of structures C and D, it can be found that when the depth of the features (the number of blocks) are the same, the better feature extraction effect can be obtained by increasing the number of convolutions in the block. Combined with the observations of structures B and D, it was found that when the number of blocks and convolutional layers were the same, increasing the number of feature extractions and the parameters of convolutional layers by increasing the number of channels could enhance the optimization effect of each iteration of the model. Finally, compared with the combination of structures A and C, it could be seen that, when the total number of convolution layers was constant, the time spent on each iteration of the model was basically not affected by the number of blocks, but the convergence effect of each iteration of the model was significantly enhanced.

In this experiment, we judged the training performance of the model by combining the convergence effect and the time spent in each iteration and selected the optimal structure of the convolution part. Finally, we chose VGGD as the optimal structure in the recognition–verification mechanism to conduct experiments on online SLR.

#### 6.1.2. Adaptive Average Pooling Layer

In the output of the convolution part, besides deep sign language features, it also contained a lot of redundant information. The use of an adaptive average pooling layer can help the network automatically select feature parameters, reduce the number of FCs, and improve the performance of the network. In this experiment, we discuss the influence of an adaptive average pooling layer on the network performance through comparative experiments. The experimental results on whether to add an adaptive average pooling layer to the output position of the network convolution section are shown in Figure 18.

Through Figure 18a, it can be seen that after using an adaptive average pooling layer, the convergence speed of the network increased to a certain extent, and at the same time, the fluctuation effect of the network also decreased. By analyzing Figure 18b, we can see that the accuracy rate of the network with the adaptive average pooling layer increased faster and the final accuracy rate was higher. The analysis above shows that the automatic feature selection and the classification function of replacing the FC using an adaptive average pooling layer achieved good results. As can be seen from Figure 18c,d, the training time and testing time of the model were reduced due to the reduction in the number of FCs. Because of the good effect in the feature selection of sign language data, the adaptive average pooling he adaptive average pooling layer was added to the classification model in the online SLR.

#### 6.1.3. The Activation Function

The comparison of the experimental results using ReLU and Leaky ReLU activation functions is shown in Figure 19. As can be seen from Figure 19a,b, there were significant differences in the iteration effect and the accuracy rate between the two activation functions in the network training process. Sign language data in this experiment had strong fluctuation characteristics, and the numerical values changed frequently between positive and negative numbers. As the ReLU activation function had a serious sparsity when its input value was negative, it was difficult to converge to good effect in the training process. However, a Leaky ReLU activation function can overcome this problem. With a Leaky ReLU activation function, the model can be trained well and achieve a high-precision classification of sign language data. As can be seen in Figure 19c,d, the training and testing speed with the Leaky ReLU activation function was slightly higher than those with ReLU, but there was no significant difference between the two. Because of its good training effect and high accuracy rate, Leaky ReLU was applied to the classification model.

#### 6.1.4. Batch Normalization

The BN renderings of the classification model training are shown in Figure 20:

The internal covariate shift problem can be overcome by BN, which greatly reduces the network fluctuation in the training process and ensures the stability and rapidity of model training. In addition, BN can standardize the different dimensions of sign language features extracted from each convolution layer, and then make it subject to the same distribution, which makes iteration of model parameters easier. As can be seen from Figure 20a, the fluctuation of loss in the networks with BN was significantly reduced and faster than that without BN. From Figure 20b, it can be concluded that BN was helpful in achieving a higher accuracy rate. It can be seen from Figure 20c,d that the training and testing time increased after using batch normalization. However, compared with the advantages of this operation for model training, this disadvantage can be ignored. Because of its good training effect, BN was applied to the classification model.

### 6.2. Verification Model

In the verification model, the training speed and the distinction between different types of data pairs were selected to measure the quality of the model. Faster training speed can ensure that developers and users of a sign language recognition system get a new model in less time. Better distinction of data pairs can ensure that the verification model has higher reliability.

#### 6.2.1. Loss Function

The experimental results using different loss functions are shown in Figure 21. Because of its smooth function characteristics, the sigmoid modification function will not cause the sparsity problem in the contrastive loss function. As can be seen from Figure 21a, in the process of training, the fluctuating range of the contrastive loss function curve was higher. The sigmoid modification function not only did not have severe fluctuation, but also had a fast convergence speed. Experimental results showed that using the sigmoid modification function as the activation function for network training can effectively overcome the sparsity problem in the contrastive loss function. As can be seen from Figure 21b,c, both activation functions can be used to distinguish the same and different data pairs of tags. Because of the good network training performance, the sigmoid modification function was used as the loss function of the verification model in online recognition.

#### 6.2.2. Model Initialization

The convolution part with the convolution parameters of the classification model can effectively extract sign language features. Therefore, using these parameters to initialize the verification model can help the training process find the parameters with good performance faster. The experimental results using different model initialization methods are shown in Figure 22. In the graph, it can be seen that using the convolution parameters of the classification model to initialize the feature extraction part of the network can significantly accelerate the training speed of the network. In the comparison of test results, when distinguishing whether the labels of the data pairs were the same, the effect of parameter initialization using only normal distribution was not as good as that with the classification model parameters. Because of the faster training speed and the distinction between different types of data pairs, the parameters of the convolution part of the classification model were used to initialize the parameters of the verification model in online recognition.

#### 6.2.3. Batch Size

The experimental results of training using different batch sizes are shown in Figure 23. It can be seen from Figure 23a that the fluctuation amplitude of loss at batch size 128 was far larger than that of the other two cases, and the gradient descent rate was the slowest. Thus, we can conclude that the change in the batch size can affect the fluctuation of the gradient descent. Larger batch size makes the distribution of the batch data closer to the distribution of total samples, making the distribution difference between batches smaller, thus making the fluctuation of loss smaller. When the batch size was 512, the fluctuation range of loss was the smallest. However, after a certain number of iterations, the training situation of the batch size with 256 and 512 was basically similar. A large batch size will cause problems. When the batch size is doubled, the number of iterations of the same epoch will also be reduced by one time. It can be seen from Figure 23a that under the same epoch, the model training effect of a batch size of 512 was not superior to that of a batch size with 256. It can also be seen from Figure 23b,c that when the batch size was 256 and 512, it was better to distinguish whether the data were the same with labels. By comparing graphs Figure 23c,d the discrimination degree of different data pairs was similar when the batch size was 256 and 128. The batch size of the verification model was 256 in online recognition because of the better training effect and the distinction between different types of data pairs.

### 6.3. Online Sign Language Recognition

In this section, we analyze some important experimental results of the recognition–verification mechanism and the segmentation–recognition mechanism. In the recognition–verification mechanism, the parameters and experimental methods of the classification model and verification model are shown in Section 5.3.2. When the online sign language data passes the classification model and verification model, it needs to be compared with the threshold to determine whether the recognition was successful. The formula of threshold is chosen as shown in Equations (4) and (5) in Section 4.3, using 2.5 times the 80% quantile as the threshold of this class. Threshold selection results for different categories are shown in Table 4.

As can be seen from the table, the number of thresholds distributed in the 14–18 range was 78, accounting for 90.70%. According to experience, when the threshold is less than 14, it is difficult to recognize the category correctly. When the threshold is greater than 18, misrecognition often occurs. Thus, the range of threshold was limited to 14–18. When the threshold was greater than 18, we replaced it with 18, and when it was less than 14, we replaced it with 14.

In the segmentation–recognition mechanism, referring to the data flow in Figure 3, the corresponding experimental steps have been given in Section 5.3.3. In this mechanism, the segmentation method uses the energy of sEMG to compare with the threshold to determine the start and end positions of the effective region in SLR. The schematic diagram of the piecewise process is shown in Figure 24.

In Figure 24, we can see that in the case of the weak action interference, most of the energy of the sEMG in the effective region of sign language was greater than the threshold, while most of the energy of sEMG in the blank region was less than the threshold in the case of weak action interference. Therefore, with less interference, the start and end positions of sign language can be determined based on this threshold. In this mechanism, we used SVM to classify sign language. The average and variance of the accuracy rate were 93.26% and 10.38%^2^ by 10-fold cross validation. Therefore, it can be concluded that the trained SVM can achieve good classification results for offline sign language data under the given feature types in this experiment.

In the experiment of online SLR, the pre-collected online sign language data were fed into the recognition–verification mechanism and the segmentation–recognition mechanism, and the experimental results of the two mechanisms were compared. The results of the online SLR experiment are shown in Figure 24.

As is shown in Figure 25, we can find that the Siamese network in the recognition–verification mechanism has a strong ability to judge the misrecognition results of the classification model. Compared with the segmentation–recognition mechanism, the misrecognition frequency of “insert”, “misalignment”, and “substitute” in the recognition–verification mechanism decreases. Because the recognition–verification mechanism uses sliding windows to pull data from the original sign language signals, there was misrecognition of “repeat” in the experimental results of the recognition–verification mechanism that did not exist in the segmentation–recognition mechanism. At the same time, the misjudgment of the correct results by the recognition–verification mechanism also made the misrecognition frequency of “delete” slightly increased compared with the segmentation–recognition mechanism. However, the error frequency of “correct” in the recognition–verification mechanism was more than twice that of the segmentation–recognition mechanism.

Because of the different operation schemes of the two mechanisms, the calculation methods for the average time of the two mechanisms were also different. The recognition–verification mechanism uses sliding windows to recognize online data. In the process of processing online datasets, the number of sliding windows pulling data was 15,263 times, and the total time was 117 s. The average processing time for each sliding window was 7.67 ms. In the segmentation–recognition mechanism, 394 effective regions of sign language were recognized. This mechanism takes 13 s to process the whole online dataset. For each online data segment recognized, the segmentation–recognition mechanism spent an average of 32.99 milliseconds on sign language recognition. By comparing the average time, it can be concluded that the delay of the two mechanisms in online SLR was not obvious, and the recognition–verification mechanism had faster recognition speed than the segmentation–recognition mechanism.

Therefore, on the whole, the performance of the recognition–verification mechanism was better than that of the segmentation–recognition mechanism.

## 7. Conclusions and Future Work

In this paper, our main research content was online SLR using sEMG sensors and IMU. Before putting forward our experimental scheme, we analyzed the possible difficulties in online SLR, and classified basic misrecognition into five types: “insertion”, “misalignment”, “repeat”, “substitute”, and “delete”. After that, we introduced the sign language database, based on which, we built a recognition–verification mechanism for online SLR. In this mechanism, a CNN based on VGG architecture was used as the classification model, and the model was optimized. The verification model used the CNN-based Siamese network, and optimized the initialization of the model and the selection of batch size, and proposed the Sigmoid correction function as the loss function for network training. After determining the relevant configuration of the recognition–verification mechanism, in order to verify the effectiveness of our method, we designed an online SLR experiment. In this experiment, we saved a section of the online sign language data collected in advance, sending the data to the recognition–verification mechanism for processing and analyzing results, and compared it with the traditional segmentation–recognition mechanism. By analyzing the experimental results, the effectiveness of the recognition–verification mechanism was verified.

Our study provides a reference method for online SLR, but more work needs to be done. In order to apply online SLR to reality, we need a larger sign language dataset that includes more sign language types. Based on more kinds of sign language, whether this method has good performance needs to be further verified. The current method to obtain data is relatively inefficient, as each category of sign language contains relatively few samples. In order to make the SLR scheme more robust online, the dataset needs to include more samples that can be seen as the same kind of sign language, and it is necessary to research a kind of sign language data generation method to expand the dataset. In the recognition–verification mechanism, the calculation of reference vectors and thresholds depends on the current dataset. If data samples are added or deleted in the dataset, reference vectors and thresholds need to be recalculated. Therefore, it is necessary to design an excellent reference vector and threshold updating method to save computational effort and ensure that the system can be optimized steadily. In addition, our research work of SLR is on the level of words now. In the future work, we can use language model to translate the recognized sign words into sentences for output. In addition, we can also try to use end-to-end method to recognize sign language at sentence level. In this way, sign language recognition has greater practical value in practical application. 

## Figures and Tables

**Figure 1 sensors-19-02495-f001:**
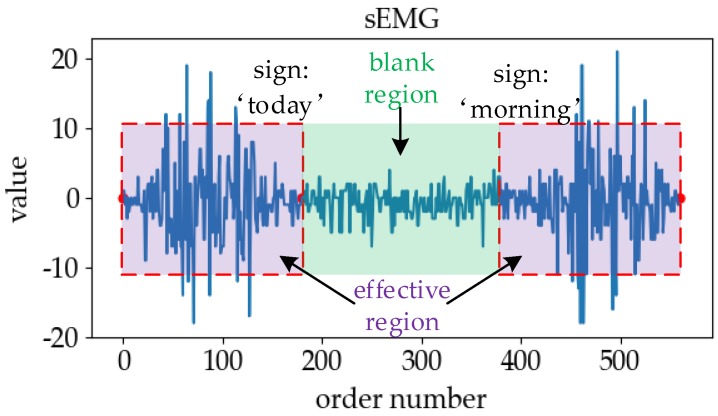
A segment of surface electromyography (sEMG) signal containing two effective regions and one blank region. The purple area in the red box represents the effective region of sign language action, while the green area represents the blank region. The left box represents the sign language of “today” and the right box is “morning”.

**Figure 2 sensors-19-02495-f002:**
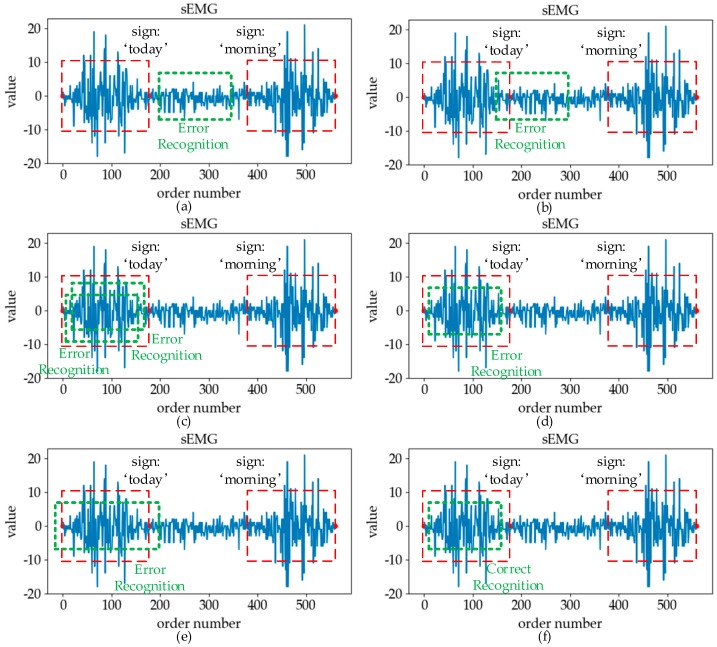
Situations in online sign language recognition (SLR): (**a**) insert; (**b**) misalignment; (**c**) repeat; (**d**) substitute; (**e**) delete; (**f**) correct. The red box indicates the effective region of sign language action, and the green box indicates the position of the recognition results, including correct recognition and misrecognition. (**a**–**e**) are misrecognition in online SLR and (**f**) is correct recognition.

**Figure 3 sensors-19-02495-f003:**
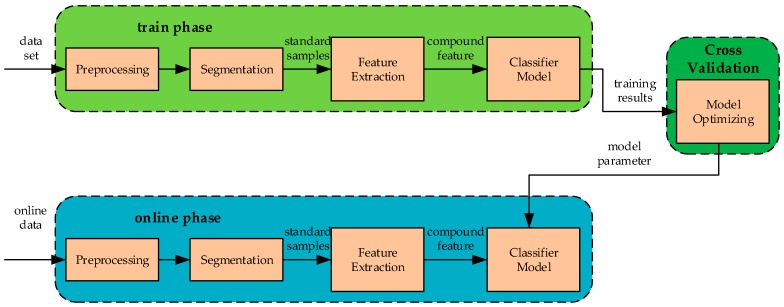
Segmentation–recognition mechanism signal flowchart. There are two parts in this flowchart: offline data processing with model training and online data processing with SLR.

**Figure 4 sensors-19-02495-f004:**
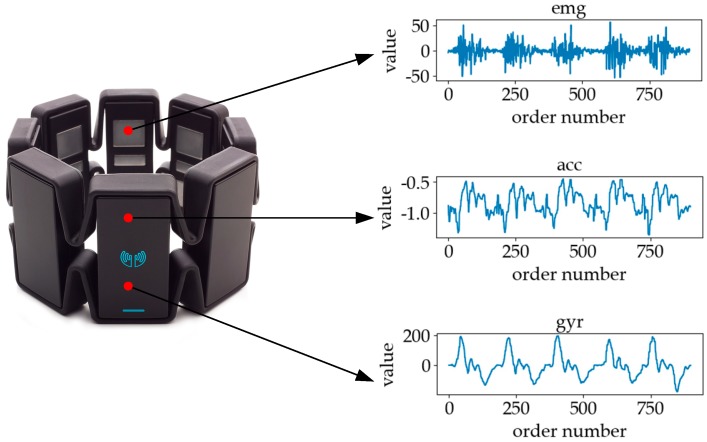
The wearable device we used to collect the sign language signals data including sEMG, accelerometer (ACC), and gyroscope (GYR).

**Figure 5 sensors-19-02495-f005:**
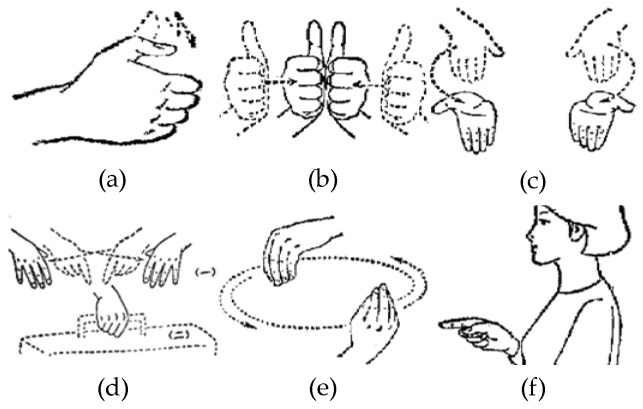
Sign language schematic. The figure shows six classes of sign language extracted from the database: (**a**) sign: “thanks”; (**b**) sign: “friend”; (**c**) sign: “what”; (**d**) sign: “luggage”; (**e**) sign: “communicate”; (**f**) sign: “you”.

**Figure 6 sensors-19-02495-f006:**
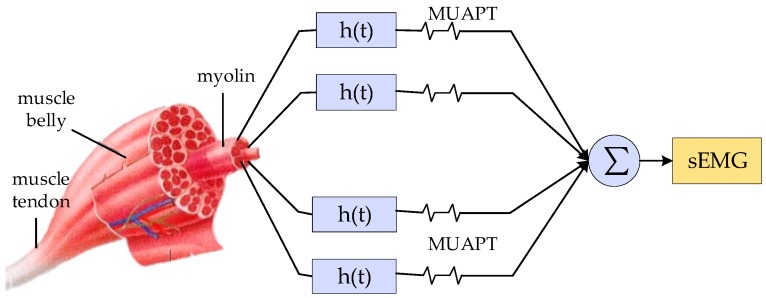
Schematic diagram of the generation principle of sEMG. On the left is the skeletal muscle composed of muscle tendons and belly. The basic unit in the belly is the muscle myolin [12]. Each muscle myolin is an independent unit of kinetic energy and structure. The action potential *h*(*t*) produced by tens of thousands of contraction changes of muscle myolins is transmitted to the skin surface and superimposed in time and space to form sEMG.

**Figure 7 sensors-19-02495-f007:**
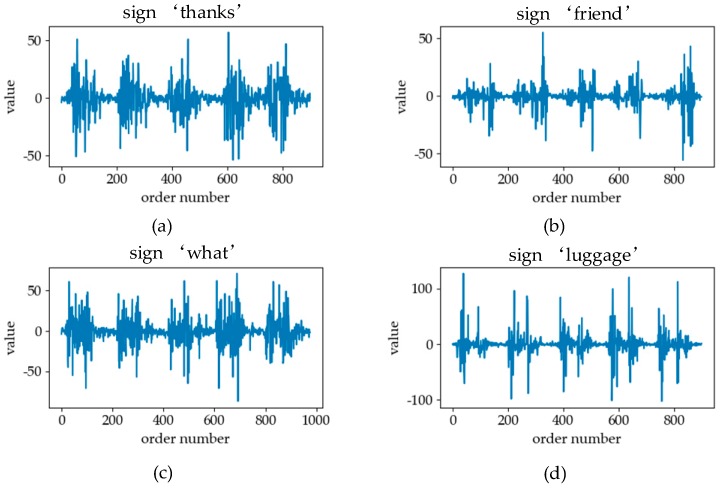
sEMG signals in different sign languages: (**a**) sign: “thanks”; (**b**) sign: “friend”; (**c**) sign: “what”; (**d**) sign: “luggage”. Each sign language signal contains five samples.

**Figure 8 sensors-19-02495-f008:**
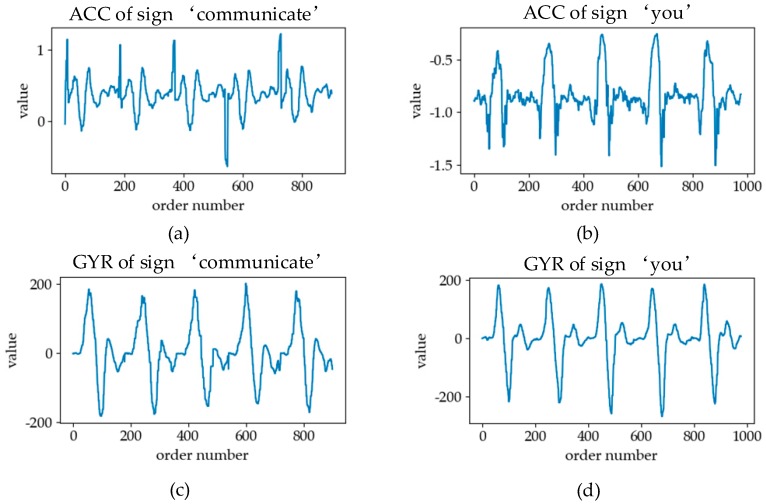
ACC and GYR signals in different sign languages: (**a**,**b**) are ACC signals of the signs “communication” and “you”; (**c**,**d**) are GYR signals of signs “communication” and “you”.

**Figure 9 sensors-19-02495-f009:**
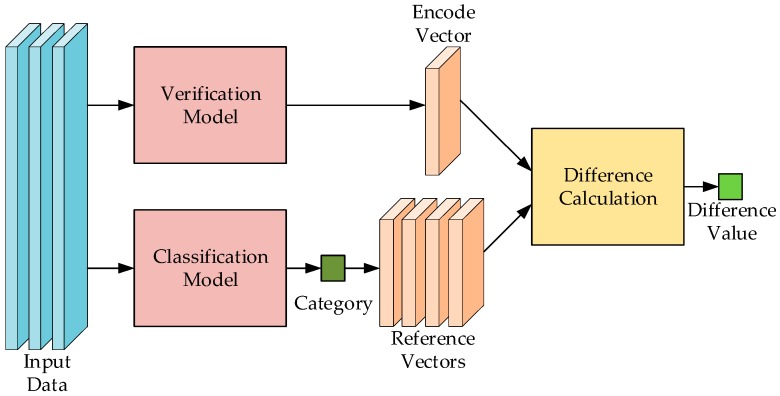
Data flow chart of the recognition–verification mechanism. The input of this mechanism is the sign language data pulled by the sliding window, and the output is the recognition category and the difference. The classification model, verification model, and difference calculation in the figure constitute the main body of the mechanism.

**Figure 10 sensors-19-02495-f010:**
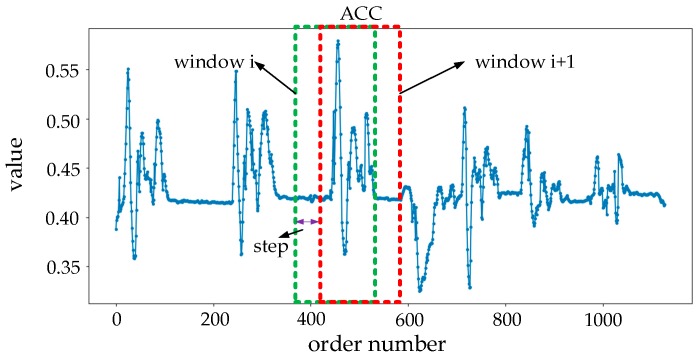
Schematic diagram of the sliding window. The two boxes in the figure are two adjacent positions during the moving process of the sliding window, the green window is the first position, and the red window is the last one during the moving process.

**Figure 11 sensors-19-02495-f011:**
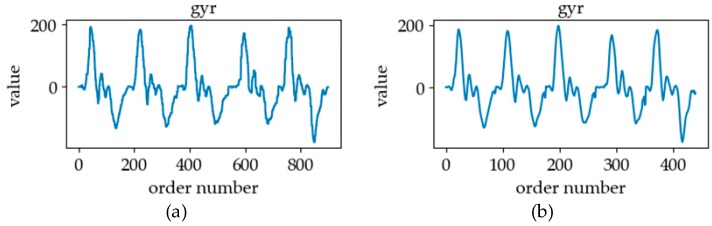
Smoothing of the GYR signal. (**a**) The original GYR signal and (**b**) the smoothed signal.

**Figure 12 sensors-19-02495-f012:**
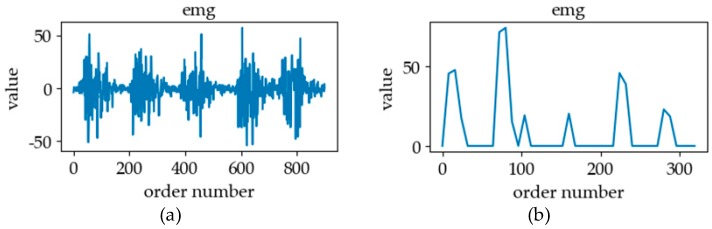
Wavelet variation of the sEMG signal. (**a**) the original sEMG image and (**b**) the transformed image.

**Figure 13 sensors-19-02495-f013:**
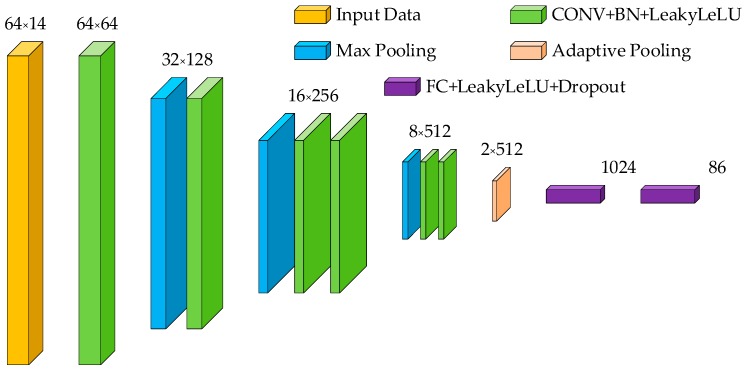
Network structure of the classification model for SLR. This model uses VGG architecture to classify sign language. The input of the model was sign language samples of 14 channels, and the output is a one-hot code vector which the dimension is 86.

**Figure 14 sensors-19-02495-f014:**
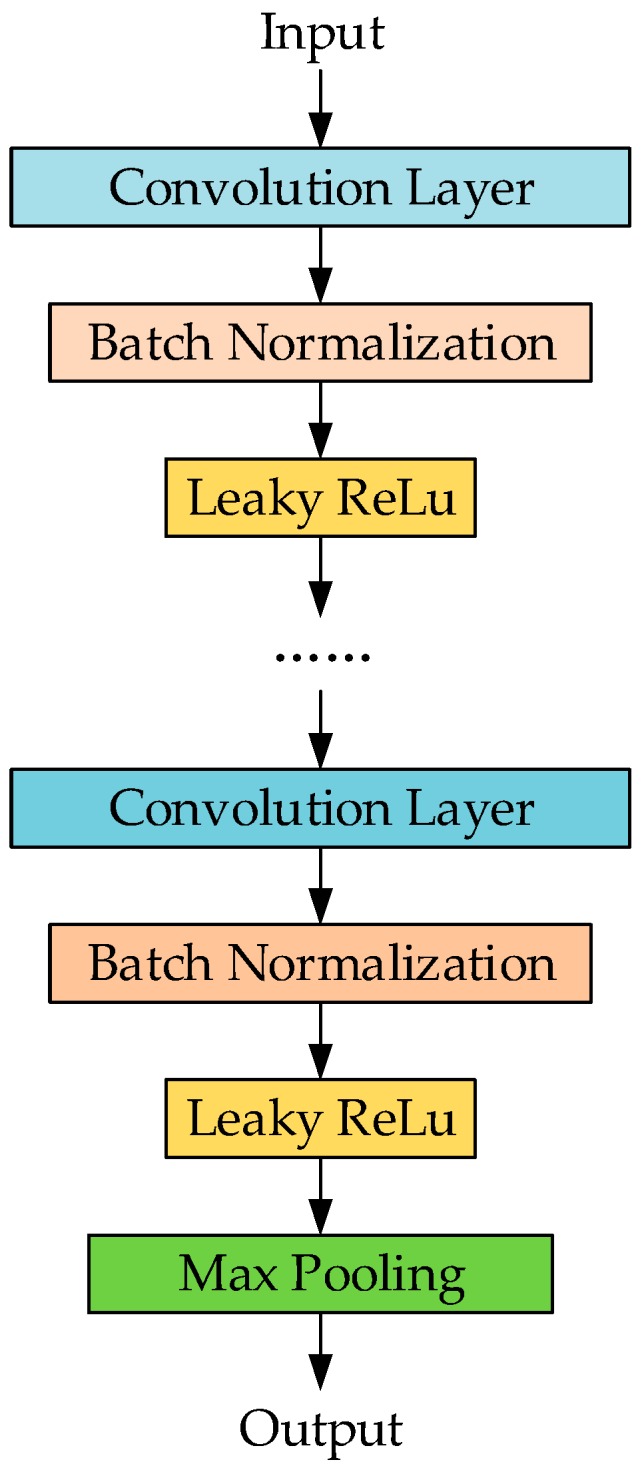
Block structure diagram. Each block is composed of multiple convolutional layers, batch standardization, and activation functions. Before the output of the block, feature filtering and data compression are carried out through the pooling layer.

**Figure 15 sensors-19-02495-f015:**
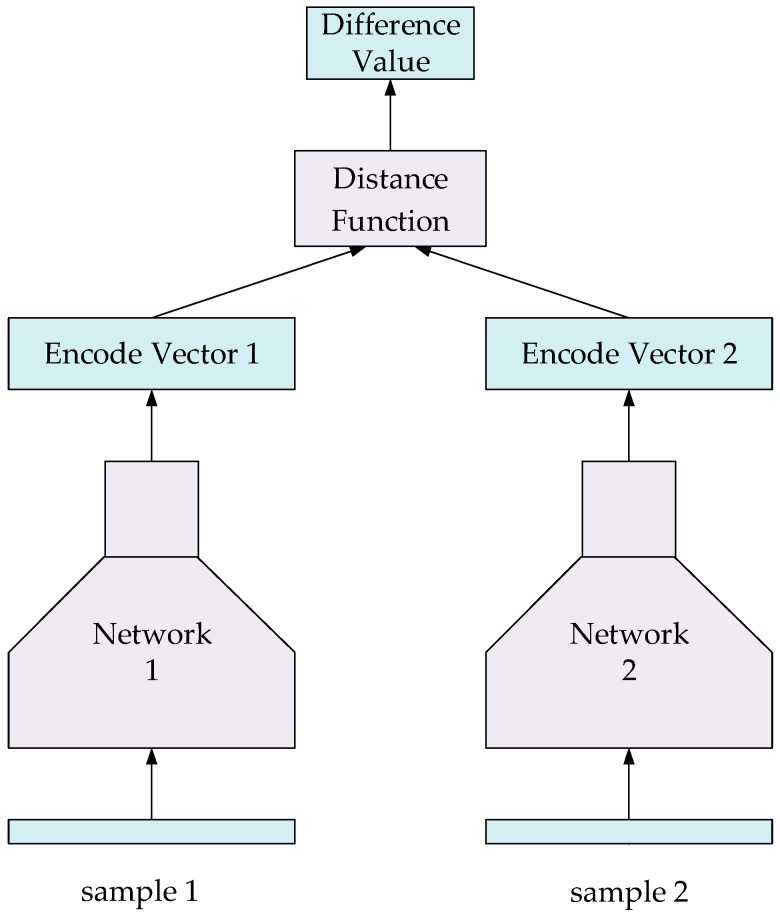
Structural diagram of the Siamese network. The network consists of two coding networks with common parameters and a distance function. The input of the network is a pair of sign language samples, and the output of the network is to measure the distance between the two samples.

**Figure 16 sensors-19-02495-f016:**
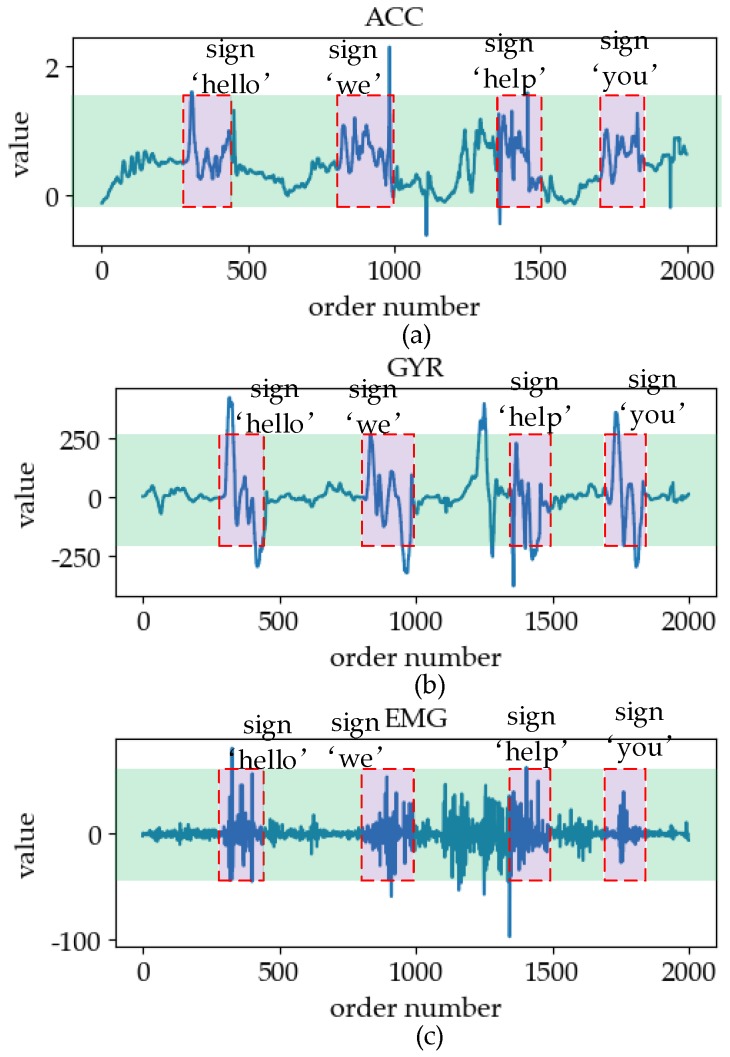
sEMG signal of online sign language data. Two-thousand data points for the whole online sign language signal intercepted are depicted in the figure, wherein the red box is the effective region of sign language data.

**Figure 17 sensors-19-02495-f017:**
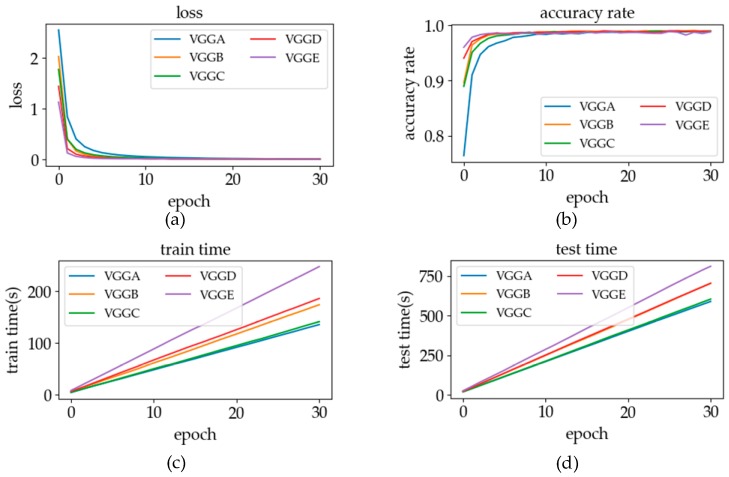
Experimental results demonstration of the convolutional network structure. The graphs show the changes of the five network structures with an epoch under one certain assessment index with epochs during the training process. Each epoch contained 601 iterative updates of the network parameters. (**a**) The change curve of loss with epoch; (**b**) the change curve of the accuracy rate with epoch; (**c**) the change curve of training time with epoch; (**d**) the change curve of test time with epoch.

**Figure 18 sensors-19-02495-f018:**
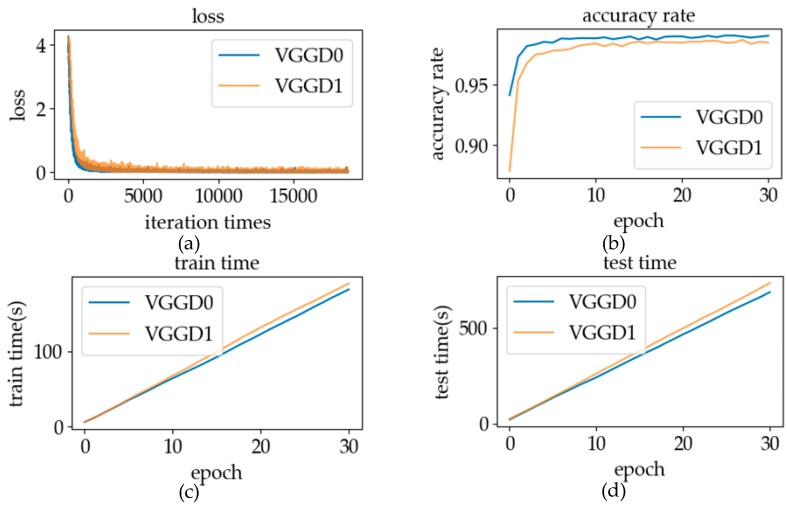
The experimental results for the adaptive average pooling layer. (**a**) The change curve of loss with iterations. (**b**) The change curve of the accuracy rate with epoch. (**c**) The change curve of training time with epoch. (**d**) The change curve of test time with epoch. The VGGD1 represents the situations without an adaptive average pooling layer and the VGGD0 represents the situations after using an adaptive average pooling layer.

**Figure 19 sensors-19-02495-f019:**
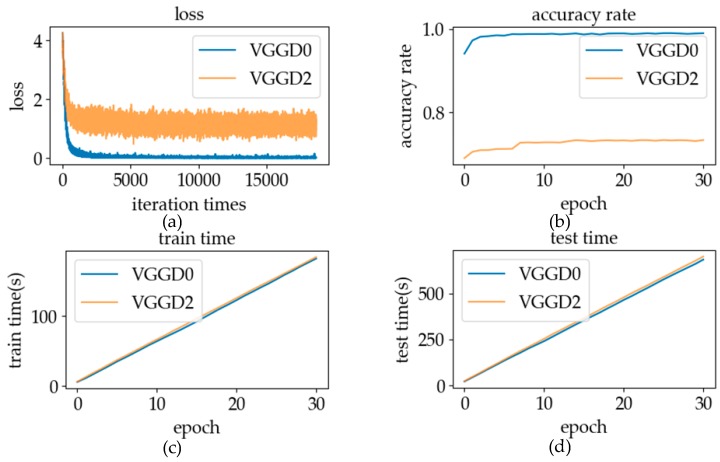
The compared experimental results of two different activation functions. Each subgraph in this diagram has exactly the same meaning as in Figure 18. The VGGD2 represents the use of the ReLU activation function, and the VGGD0 represents the use of the Leaky ReLU.

**Figure 20 sensors-19-02495-f020:**
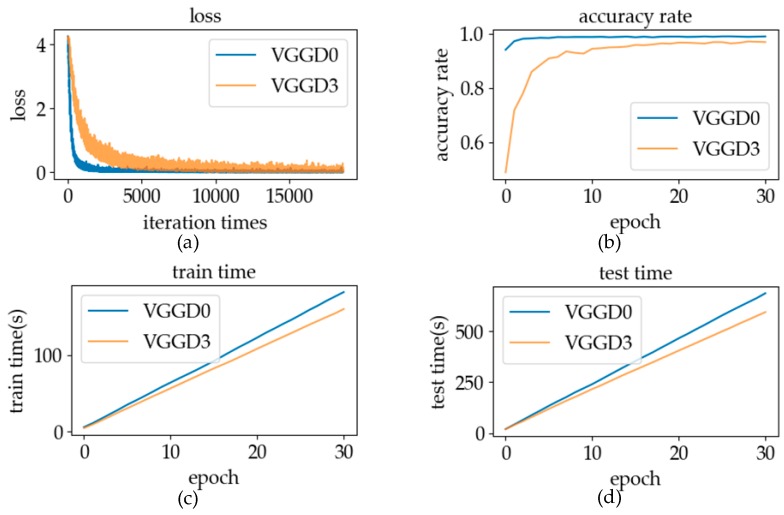
The effect diagram of BN. Each subgraph in this diagram has exactly the same meaning as in Figure 18. Wherein the VGGD0 represents the use of BN and the VGGD3 represents unused.

**Figure 21 sensors-19-02495-f021:**
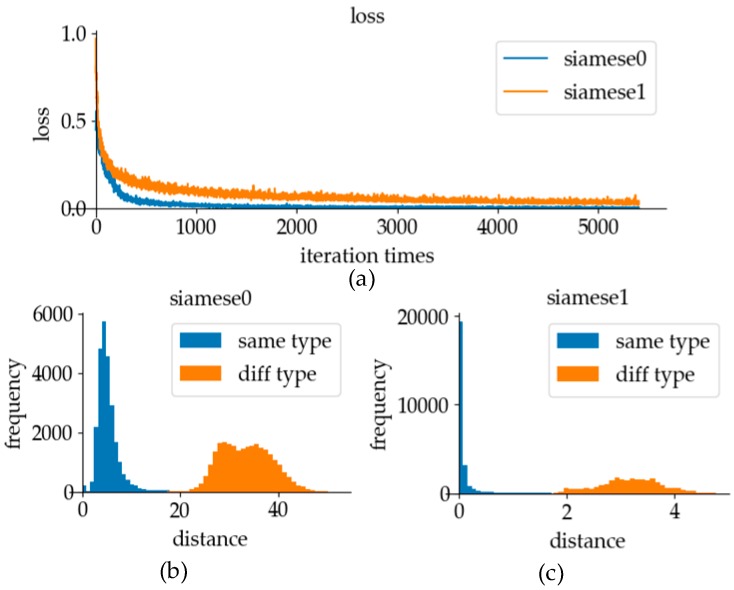
Experimental results compared with different loss functions. (**a**) The variation diagram of loss with iterations in two ways, wherein the siamese0 represents the contrastive loss used as the loss function in the training process, while the siamese1 represents the sigmoid modification function used. (**b**,**c**) Histograms of test results in two ways, where each long column represents the number of samples within the range of Euclidean distance, in which the “same type” represents the same type of data pair and the “diff type” represents the different type of data pair. (**b**) shows the test results using the sigmoid modification function. (**c**) shows the test result diagram using the contrastive loss function.

**Figure 22 sensors-19-02495-f022:**
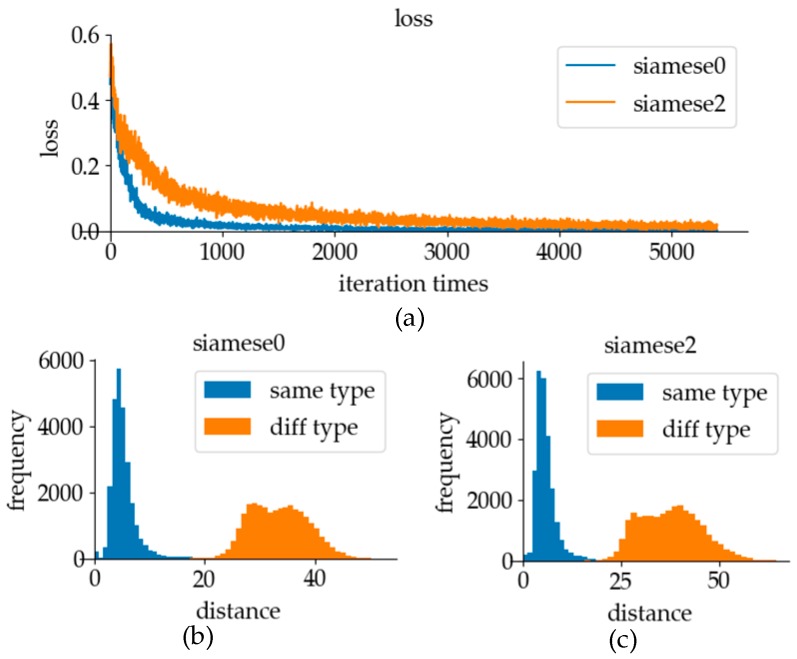
Comparison of experimental results with different initialization methods. Each of these subgraphs has the same meaning as Figure 21. (**a**) The siamese0 represents the situation under the normal distribution parameter initialization method used in the training process, and the siamese2 represents that under the parameters trained by the classification model for feature extraction. (**b**) The test result diagram initialized with the classification model parameters. (**c**) A test result diagram that only used normal distribution for parameter initialization.

**Figure 23 sensors-19-02495-f023:**
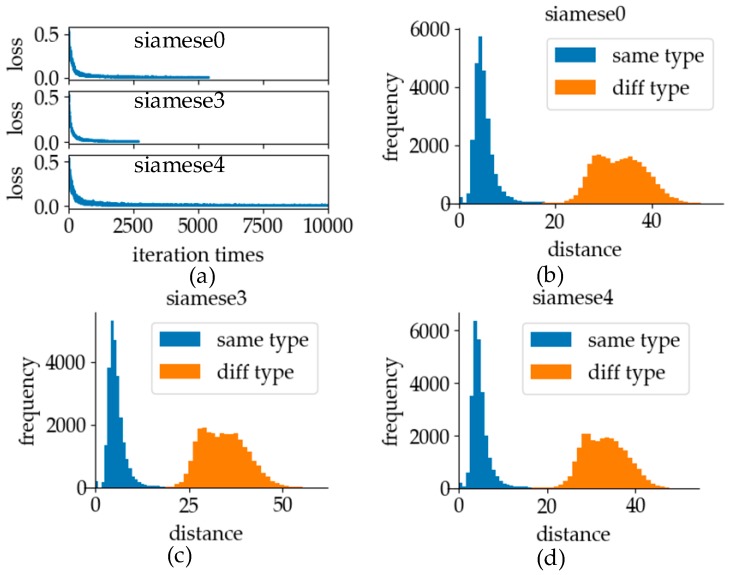
Comparison of the experimental results of network training using different batch sizes. Each of these subgraphs had the same meaning as Figure 21. (**a**) Siamese0 represents the situation of the batch size 512, siamese3 represents 256, and siamese4 represents 128. (**b**) The test results whose batch size was 256. (**c**) The test results whose batch size was 512. (**d**) The test results whose batch size was 128.

**Figure 24 sensors-19-02495-f024:**
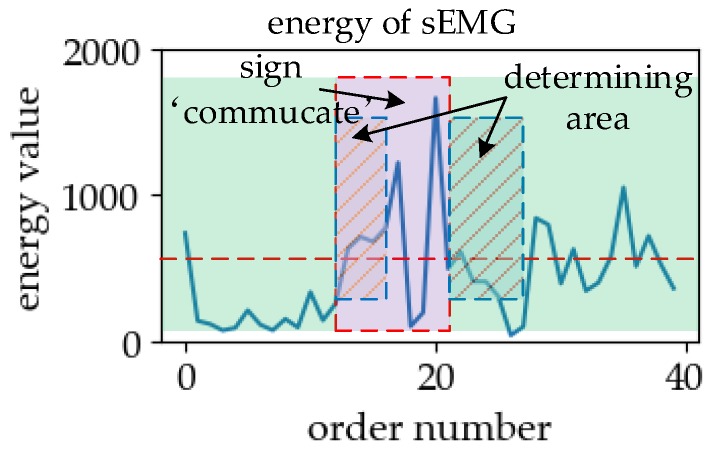
Segmentation in the segmentation–recognition mechanism. The purple part in the red box is the effective region of sign language “communication”. The green shadows are the blank region. The shaded part with oblique lines in the blue box is the determining area for the beginning and ending positions of the effective region. The red line is the contour of the energy threshold.

**Figure 25 sensors-19-02495-f025:**
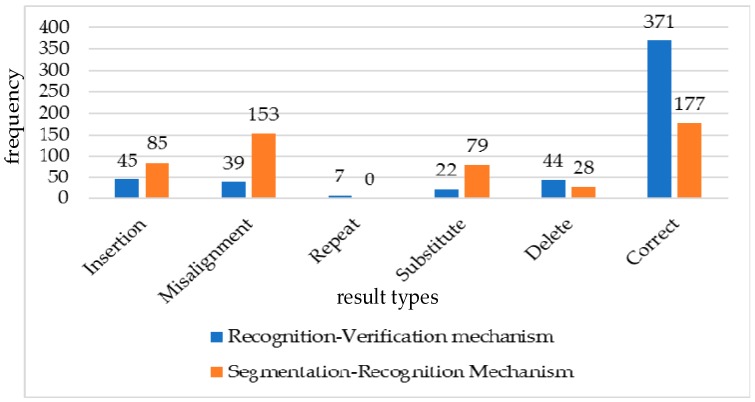
Comparison diagram of online SLR experiment results. In this figure, the *x*-axis is the result types including five types of misrecognition and a correct recognition and the *y*-axis is the frequency of each result. The orange histogram is the experimental result of the recognition–verification mechanism. The blue histogram is the experimental result of the segmentation–recognition mechanism.

**Table 1 sensors-19-02495-t001:** List of sign language categories. “[]” stands for “empty” class.

Sign Language Categories								
friend	today	think	luggage	how	good	doctor	afternoon	one
hello	egg	me	ok	where	support	lighter	morning	dusk
sky	home	airport	consign	find	we	fever	yesterday	say
evening	return	who	take off	why	Shenyang	teacher	communicate	ask
happy	go	toilet	time	ID card	help	like	how to go	do
noon	late	refund	miss	terminal	China	listen	not found	[]
thanks	meat	ticket	change	watch	Liaoning	come	tomorrow	
sorry	contact	worry	flight	key	world	love	confiscate	
it’s ok	you	how	delay	knife	Beijing	want	everybody	
eat	what	question	excuse	cigarette	charge	drink	(early) morning	

**Table 2 sensors-19-02495-t002:** Convolutional network layer configuration. Five convolutional network layer structures are defined from VGGA to VGGE.

VGGA	VGGB	VGGC	VGGD	VGGE
Conv3-64	Conv3-64	Conv3-64	Conv3-64	Conv3-64
MaxPooling				
Conv3-128	Conv3-128	Conv3-128	Conv3-128	Conv3-128
MaxPooling				
Conv3-256	Conv3-256	Conv3-256	Conv3-256	Conv3-256
Conv3-256	Conv3-256		Conv3-256	Conv3-256
MaxPooling				
	Conv3-256	Conv3-512	Conv3-512	Conv3-512
	Conv3-256		Conv3-512	Conv3-512
	MaxPooling			
				Conv3-512
				Conv3-512
				MaxPooling
APL				
FC-86				

**Table 3 sensors-19-02495-t003:** Experimental results of the convolutional network structure.

Model	Loss = 0.5			Loss = 0.1			Loss = 0.05			Loss = 0.01		
	e	AR	t	e	AR	t	e	AR	t	e	AR	t
VGGA	3	94.72	13.41	8	97.96	35.05	12	98.55	52.56	0	0	0
VGGB	2	96.44	11.27	5	98.41	28.03	7	98.55	38.8	17	98.92	94.89
VGGC	2	95.11	9.05	5	98.09	22.43	8	98.56	36.28	20	98.92	90.54
VGGD	2	97.07	12.08	4	98.4	24.18	5	98.63	30.29	14	98.83	84.98
VGGE	2	97.85	16.05	3	98.31	24.05	4	98.51	31.98	11	98.54	88.45

**Table 4 sensors-19-02495-t004:** Sign language thresholds. Thresholds for each location in the table correspond to the location in Table 1.

Sign Language Thresholds
14.90	14.38	16.17	14.06	14.53	14.87	17.55	16.33	16.35
15.23	14.58	14.42	15.06	15.42	14.38	14.83	14.93	17.26
16.11	14.92	15.27	15.77	16.56	14.58	16.38	14.82	15.14
19.03	17.28	14.62	15.25	17.24	15.23	14.27	16.46	14.12
16.47	14.23	15.30	16.17	15.29	13.49	15.63	17.26	16.37
14.67	17.93	16.88	15.19	12.99	15.16	13.22	14.62	21.99
16.33	15.39	15.98	13.60	19.03	15.29	16.15	15.76	
16.59	14.64	14.98	15.10	14.18	15.29	15.38	16.45	
14.73	14.76	16.01	14.99	14.99	15.72	14.53	14.89	
14.96	14.26	14.72	16.32	13.76	14.58	15.76	15.34

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
