# Peer review of "An Recognition–Verification Mechanism for Real-Time Chinese Sign Language Recognition Based on Multi-Information Fusion"

_sensors, 2019, doi:10.3390/s19112495_

Round 1
Reviewer 1 Report
Authors have presented a study for SLR using Myo armband that utilized sEMG, ACC and GYR signals. The study is interesting, and authors tend to focus on diversity of CNN hyperparameters that affect the accuracy rate. There are few issues that need to be attended as well before publication. 1) What does it mean by "wear the Myo armband several times ..." at page 6, line 207? How do you wear several time and collect data at a time? Quite confusing statement. 2) How the machine selects the standard reference vector from the set of vectors of all categories based on output type of model? How the selection is carried out? Under what circumstances using loss function, the standard reference vector and input vector is verified to be equally alike or otherwise? 4) What is the sliding window size? 5) Please include the figures of SL used in this study for clarification. 6) How was the 64 vector size of inputs are defined? 7) Fig. 21 is hard to comprehend, what is the x-axis and y-axis, the explanation mentioned that RVM can judge error better than SRM, but how it relates to the Fig 21, please explain in more detail. 8) The whole paper is focusing more on the hyperparamaters of CNN in determining the accuracy rate, but lack of discussion on the dataset itself. Basically, readers are quite confuse on what you trying to compare with? What are the sign signals? How the sign signals work? Will it involve orientation? Authors can provide several examples of the EMG and IMU signals corresponding to different words, and what are the library words used here for training and testing set. Are there any similarity between the SL used? If exists, does the usage of different hyperparameters can increase the detection of those words? 9) No detail explanation on the experiment setup, how experiments are conducted, what's are the average time for the whole process, average time between the words? Kindly include several figures to further illustrate the conducted experiments. 10) Mostly, authors only presented results with EMG signals, but hardly to see any results related to ACC and GYR, but authors did use those data for training and testing, so inclusion of those data for result discussion is very important as well. 11) In the introduction (literature review section), authors should also presented similar studies of using Myo and what are the different approaches used in this study. 12) In the final results, authors claimed that using RVM performed better than traditional SRM (especially in computer vision), authors should provide several studies for results comparison in detail.Author Response
Dear Review:
RC: Authors have presented a study for SLR using Myo armband that utilized sEMG, ACC and GYR signals. The study is interesting, and authors tend to focus on diversity of CNN hyperparameters that affect the accuracy rate. There are few issues that need to be attended as well before publication.
AR: First of all, we would like to thank you for your careful comments and your valuable suggestions. We try our best to take your valuable opinions into account and make the best revisions to our paper in the process of correcting our paper. It is worth mentioning that, for your convenience in reading, the revised contents of manuscript are indicated in revision mode of word, and clearly marked in the revised manuscript, such as "review 1:1)" .

Reviewer 2 Report
In the manuscript, a Recognition-Verification mechanism is proposed for sign language recognition. The paper will be reconsidered after revising/answering the followings.
(1) Please provide precise definition on FZ and AZ.
(2) In Figure 1, please mark different regions and zones for better understanding.
(3) In Figure 2(d), please indicate the S type misrecongnition clearly.
(4) How is the sliding window define?
(5) Please describe the applied encoders in the procedure.
(6) In the manuscript, although there are only five equations, the expressions are very confusing. Please revise them carefully and consistent the symbols for vectors, functions, coefficients, etc.
(7) In Eq. (1), y should be a function of x, A, B, and C. In Eq. (2), y changes to a function of x and omega. Which one is correct?
(8) In Eq. (2), why “1/2” is needed (instead of “1”)?
(9) Please explain all the new parameters in Eq. (3).
(10)In line 329-330, the authors mentioned “Combined with our experiments”, please explain.
(11)In Section 4.3.3., please give details about the variation of the training rate.
(12)In Section 4.4.3, I suggest changing “Use stage” to “Operation stage”.
(13)The explanation of the threshold value and its implementation should be expressed in formulas.
(14)Please provide an overall flowchart of the proposed method.
(15)In Eq. (4), please give detail explanation of W.
(16)In Eqs. (4) and (5), the “margin” become “m arg in” in the equations. I suggest to assign a symbol to replace the “margin”.
(17)In Section 6, please consistent “Figure”, “figure”, “fig”, …; “VGG”, “vgg”; “LOSS”, “LOss”, “loss”, etc.
(18)What is the unit of the training time and testing time?
(19)Please explain why the testing time is several times higher than the training time.
(20)Why vggD3 is presented before vggD2?
(21)In Figure 16, the accuracy rate and loss should be swapped.
(22)Why siamese3 is presented before siamese2?
(23)In Section 6.2.3., if the batch size is 128, how the result changes?
(24)Please discuss whether the required computational time can realize real-time recognition.
Author Response
Dear Review:
RC: In the manuscript, a Recognition-Verification mechanism is proposed for sign language recognition. The paper will be reconsidered after revising/answering the followings.
AR: First of all, we would like to thank you for your careful comments and your valuable suggestions. We try our best to take your valuable opinions into account and make the best revisions to our paper in the process of correcting our paper. It is worth mentioning that, for your convenience in reading, the revised contents of manuscript are indicated in revision mode of word, and clearly marked in the revised manuscript, such as "review 2: (1)" .

Reviewer 3 Report
The paper can be very hard to read, mostly because of the overuse of terms and acronyms. This makes getting through the paper very onerous. The writing style is often overly colloquial and/or too specific to this piece of work. For example, "generally people don't understand" is very colloquial. SLR is an ok acronym but when combined with NSL, SRM, etc., the paper starts to get unreadable. I've never seen the term "father-field" . in use. This seems way too colloquial for a formal paper. What is an RVM? I understand what it means in the context of this work but is that a more general term used in the literature? I have not seen it. In the captions, you use the terms SR, BR, sEMG, etc. Again, very confusing. Section 2.1 is very difficult to understand largely because of the use of too many acronyms.
It is not clear where there is novelty. It is stated that previous work focus on offline SLR and you addressed the online case. How is pre-processing performed in the offline case to allow it to execute in an online fashion? If we were to add the data-cleaning/noise recognition elements of your pipeline into a pipeline that includes classification for clean segments, would we see similar performance?
You should explain the types of interference in the online case. You make a general statement about it but don't characterize it more specifically. Is the interference additive? Is interference embedded before/after the SLR gesture? It's not clear which kind of interference you are designing for in your pipeline.
The choice of model for each task is not clear. Could a simpler model have done as good a job as the ones you proposed in the various stages of the classification pipeline?
Author Response
Dear Review:
AR: First of all, we would like to thank you for your careful comments and your valuable suggestions. We try our best to take your valuable opinions into account and make the best revisions to our paper in the process of correcting our paper. It is worth mentioning that, for your convenience in reading, the revised contents of manuscript are indicated in revision mode of word, and clearly marked in the revised manuscript, such as "review3 :1)".

Round 2
Reviewer 1 Report
Thank you for the authors responses.
1) It is suggested to remove the content regarding to 're-wear', it is common sense to take off and put-on sensor on different person, this writing will make readers more confusing and lower the quality of the content.
7) threshold calculation could be varied depending on the training dataset, thus, if new data is added or some old data is removed, the threshold has to be calculated again, it is suggested to list this as limitation in the discussion or conclusion if there is no valid solution at the current stage.
8) There are some typo error in the contents, e.g., "erroe", please carefully revised all.
9) Authors only answered partial of the questions, e.g., what SL being used, what are their similarity and differences, is the raw data itself is the input vectors?
Author Response
Dear Reviewer:
RC: Thank you for the authors responses.
AR: First of all, we are glad to receive your review again. We thank you very much for your valuable comments and suggestions. We try our best to take your valuable opinions into account and make the best revisions to our manuscript. By the way, for your convenience in reading, the revised contents of manuscript are indicated in revision mode of word, and clearly marked in the revised manuscript, such as "review1:1)" .

Reviewer 2 Report
The authors have addressed most of my previous comments. The follow-up questions are listed below.
(2.1) Regarding my previous question (23), the authors added Figure 23. In this figure,
(i) the last line of the description title '(b)' should be '(d)'
(ii) In the subplot (a), why the loss decreases when the batch size decreases from 512 to 256 but dramatically increase when the batch size decrease from 256 to 128?
(2.2) In the revised version, it is found that some descriptions and expressions are difficult to understand. The authors are suggested to polish the English for enhancing the presentation.
Author Response
Dear Reviewer:
RC: The authors have addressed most of my previous comments. The follow-up questions are listed below.
AR: First of all, we are glad to receive your review again. We thank you very much for your valuable comments and suggestions. We try our best to take your valuable opinions into account and make the best revisions to our manuscript. By the way, for your convenience in reading, the revised contents of manuscript are indicated in revision mode of word, and clearly marked in the revised manuscript, such as "review2:(2.1)" .

Round 3
Reviewer 2 Report
The authors addressed all my previous comments.